# Ego-R1: Chain-of-Tool-Thought for Ultra-Long Egocentric Video Reasoning

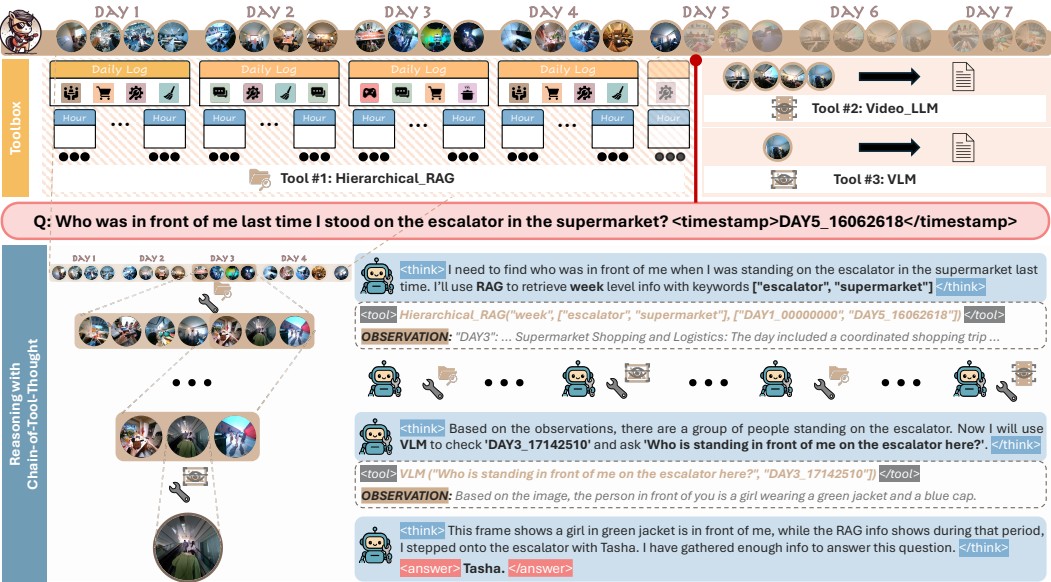

Figure 1: **Overview of Ego-R1.** In this figure, we demonstrate how the Ego-R1 Agent orchestrates specialized tools (e.g., Hierarchical_RAG, Video LLM, and VLM) to answer the question step-by-step, based on the observations and previous actions. The system effectively answers questions that require careful searching within ultra-long videos and precise analysis of frame details.

## ABSTRACT

We introduce **Ego-R1**, a novel framework for reasoning over *ultra-long* (i.e., in days and weeks) egocentric videos, which leverages a structured **Chain-of-Tool-Thought (CoTT)** process, orchestrated by an **Ego-R1 Agent** trained via reinforcement learning (RL). Inspired by human problem-solving strategies, CoTT decomposes complex reasoning into modular steps, with the RL agent invoking specific tools, one per step, to iteratively and collaboratively answer sub-questions tackling such tasks as temporal retrieval and multi-modal understanding. We design a two-stage training paradigm involving supervised finetuning (SFT) of a pretrained language model using CoTT data and RL to enable our agent to dynamically propose step-by-step tools for long-range reasoning. To facilitate training, we construct a dataset called **Ego-R1 Data**, which consists of **Ego-CoTT-25K** for SFT and **Ego-QA-4.4K** for RL. Furthermore, our Ego-R1 agent is evaluated on a newly curated week-long video QA benchmark, **Ego-R1 Bench**, which contains human-verified QA pairs from hybrid sources. Extensive results demonstrate that the dynamic, tool-augmented chain-of-thought reasoning by our Ego-R1 Agent can effectively tackle the unique challenges of understanding ultra-long egocentric videos, significantly extending the time coverage from few hours to a week.

## 1 INTRODUCTION

Egocentric videos, which capture human daily lives from a first-person perspective, are inherently long - often spanning hours to days or even weeks (Yang et al., 2025a). Understanding these videos

is crucial for supporting practical tasks such as memory recall, multi-step activity tracking, and goal monitoring (Grauman et al., 2024; Mangalam et al., 2023; Chen et al., 2024a). But the ensuing problem poses significant challenges due to the video length, multi-modality, and the need for long-horizon reasoning across diverse temporal contexts and dependencies.

Recent advances in multimodal long-context modeling have led to promising progress, extending video understanding capabilities from minutes to hours (Zhang et al., 2024a; Chen et al., 2024b; Zhang et al., 2024b; Li et al., 2024b). However, these models still face significant computational challenges and scale poorly when applied to videos of extended durations, such as those spanning a day or longer. To this end, prior works have proposed token compression (Song et al., 2024; Shen et al., 2024; Weng et al., 2024; Shu et al., 2024; Jiang et al., 2025) or sampling-based strategies that reframe video understanding as a temporal retrieval task (Qu et al., 2025; Ye et al., 2025). Nevertheless, these approaches risk missing key events due to the lossy representations or incomplete temporal localization. Another line of works, commonly referred to as video agents, leverages external language models as high-level control and reasoning entities to call specialized vision modules/tools for video reasoning (Wang et al., 2024a; Ye et al., 2025; Zhi et al., 2025). While allowing more flexible and more granular perception, these approaches still rely on predefined reasoning pipelines or fixed-order tool invocations, limiting the video lengths they can handle, i.e., up to hour-long.

To address these limitations, we propose **Ego-R1**, a novel framework that leverages fine-tuned large language models (LLMs) and reinforcement learning (RL) for *dynamic* tool-driven reasoning of *ultra-long* (i.e., in days and weeks) egocentric videos. The key distinction from prior video agents (Wang et al., 2024a; Ye et al., 2025; Zhi et al., 2025) designed for long-form video understanding is the dynamic tool calling of our **Ego-R1 Agent**, which iteratively processes both visual information and contexts to select and execute specialized perception tools *on demand*, based solely on previously observed content and thought to preceding sub-questions. We call such a video understanding paradigm **Chain-of-Tool-Thought (CoTT)** reasoning. Furthermore, unlike traditional methods that either feed the entire video to the model or select a subset of the frames, Ego-R1 utilizes a structured toolkit for perception which consist of three core modules designed specifically to facilitate efficient temporal retrieval and detailed visual comprehension. For retrieval, *Hierarchical Retrieval-Augmented Generation (H-RAG)* extracts timestamped, question-relevant information in the language space. For visual analysis, a specialized *Video-LLM* interprets localized visual contexts, while a general-purpose *Vision-Language Model (VLM)* extracts fine-grained visual details. by an orchestrating LLM trained through RL, Ego-R1 enables scalable, step-by-step compositional reasoning over ultra-long videos. The modular design of our framework enables easy integration with a wide range of state-of-the-art visual understanding models, allowing the visual perception components, i.e., the Video-LLM and VLM, to seamlessly integrate into our framework.

To facilitate the training of Ego-R1, which consists of a supervised fine-tuning (SFT) stage and an RL stage, we construct **Ego-R1 Data**, a comprehensive hybrid-source dataset consists of 25K CoTT reasoning traces and 4.4K annotated question-answer (QA) instances to support SFT of a pretrained LLM and RL training of our Ego-R1 agent, respectively. Each task within the dataset requires reasoning over substantial temporal spans, with an average of 7.42 tool-calling steps per task. Additionally, we introduce **Ego-R1 Bench**, a carefully curated evaluation framework consisting of week-long egocentric videos that combine human-annotated and post-verified synthetic data, designed specifically to assess long-horizon reasoning capabilities in the egocentric setting.

Extensive experiments across diverse long-video benchmarks demonstrate that the dynamic, tool-augmented chain-of-thought reasoning by our Ego-R1 Agent can effectively tackle the unique challenges of understanding ultra-long egocentric videos, significantly extending the time coverage from few hours to a week. We also perform ablation studies to replace the visual modules in Ego-R1 to showcase that our framework is customized to integrate current MLLMs scope, validating our method's robustness and generalization. At last, while we focus on egocentric long videos in this work, we show that our framework generalizes well in the exocentric setting as well.

## 2 RELATED WORK

**Egocentric long video understanding.** Existing large-scale egocentric datasets such as Ego4D (Grauman et al., 2022), EgoExo4D (Grauman et al., 2024), Epic-Kitchens (Damen et al., 2018), and HD-Epic (Perrett et al., 2025) have established comprehensive benchmarks (Mangalam et al., 2023; Cheng et al., 2024; Chen et al., 2023) focused on temporal understanding of daily activities, object

interactions, and episodic memory tasks (Song et al., 2023; Li et al., 2025b; Di & Xie, 2024; Tang et al., 2023; Rodin et al., 2024; Goletto et al., 2024). Despite their breadth, these benchmarks typically cover only short video durations on the order of minutes. Recent extensions have scaled to hours (Chandrasegaran et al., 2024; Ye et al., 2025), yet challenges such as multi-person interactions and cross-day behavioral reasoning remain underexplored. More recently, EgoLife (Yang et al., 2025a) introduced a week-long egocentric dataset capturing continuous daily activities, extending egocentric research to longer temporal horizons. Building on this, we propose Ego-R1 Bench, a benchmark for ultra-long egocentric video reasoning that emphasizes logical reasoning across week-scale egocentric recordings.

Existing approaches face critical limitations: proprietary models (Achiam et al., 2023; Team et al., 2024) and some MLLMs (Li et al., 2024a; Bai et al., 2025) usually process videos as unified inputs, which becomes prohibitively token-intensive for hour-long videos; general frame sampling approaches (Zhang et al., 2024a;b; Wang et al., 2025b; Liu et al., 2024b; 2025) cannot guarantee question-relevant frames selection; and video agents (Wang et al., 2024a; Ye et al., 2025; Song et al., 2024; Wang et al., 2023b; 2024b) often analyze frames in isolation, overlooking narrative structure and temporal dynamics. Retrieval-augmented generation (RAG) has emerged as a promising alternative (Luo et al., 2024; Xu et al., 2024b), yet existing methods lack contextual specificity for multi-day egocentric videos, where routines and social dynamics evolve over time. To address this, Ego-R1 leverages hierarchical RAG with multi-step reasoning, enabling a comprehensive understanding of evolving contexts beyond the single-step reasoning of Video-R1 (Feng et al., 2025b). Details of frameworks comparison and qualitative exemplars for reasoning types are shown in Table **??** and Fig. 5 respectively.

**Multimodal agentic tool-use.** Agentic systems with Tool-Integrated Reasoning (TIR) effectively enhance LLMs' complex problem-solving and reasoning capabilities (Parisi et al., 2022; Yao et al., 2023), particularly in mathematical domains (Gou et al., 2023; Wang et al., 2023a; Zhou et al., 2023; Yang et al., 2024a) through search engines (Jin et al., 2025; Zheng et al., 2025) and code interpreters (Yao et al., 2025; Liao et al., 2024; Yang et al., 2024b). For training paradigms in tool-integrated learning, RL has emerged as a promising approach offering more scalable and generalizable tool utilization strategies (Qian et al., 2025; Li et al., 2025a; Feng et al., 2025a; Wang et al., 2025a), compared to traditional SFT (Schick et al., 2023; Qin et al., 2023). Recent research has extended tool-augmented foundation models to multimodal domains, exploring the integration of diverse tool-use for visual reasoning tasks (Ke et al., 2025; Deng et al., 2025; Su et al., 2025; Maaz et al., 2024; Zhi et al., 2025; Ma et al., 2024). These initial efforts leverage specialized visual perception modules (Wang et al., 2024a; Fan et al., 2025), to enhance grounded and context-aware reasoning in complex visual environments (Chen et al., 2024b; Liu et al., 2024a). Coinciding with OpenAI's o3 (OpenAI, 2025b), Ego-R1 Agent employs dynamic tool-calling mechanisms, enabling multi-step reasoning and contextual tool selection, determining the appropriate tool for optimal problem-solving.

**CoT reasoning.** Chain-of-Thought (CoT) reasoning (Wei et al., 2022) has emerged as a fundamental mechanism to enhance the reasoning capabilities of both LLM and VLM (Xu et al., 2024a; Thawakar et al., 2025; Wang & Zhang, 2025; Wu et al., 2025; Liu et al., 2024c). RL-based reasoning approaches further require high-quality CoT samples to advance multimodal reasoning capabilities (Huang et al., 2025; Yang et al., 2025b; Dong et al., 2024; Zhang et al., 2025). However, existing datasets lack adequate, high-quality CoT annotations for long video understanding tasks. To fill this gap, we introduce Ego-CoTT-25K, featuring CoT reasoning with dynamic tool-calling capabilities.

## 3 EGOCENTRIC LONG VIDEO REASONING VIA DYNAMIC TOOL-CALLING

The egocentric long-video reasoning task represents a crucial frontier beyond understanding, as first-person perspectives capture complex, temporally interdependent human behaviors over ultra-long durations. Actions that occur many hours or even days apart may be guided by consistent personal strategies and habits; thus, correctly answering a query often relies on recognizing enduring human traits and linking them to cues dispersed across the entire timeline. This requires the models to therefore maintain long-range temporal dependencies, identify subtle evidence in earlier segments, and reason about the actor's underlying preferences to generate dynamic context-aware solutions. Although recent MLLMs demonstrate promising performance in general video understanding, they still struggle in answering questions in truly long-context videos with extended temporal relationships.

This underscores the importance of egocentric long-video reasoning as a fundamental challenge for multimodal systems. In this section, we introduce Ego-R1, a novel framework that unifies visual content comprehension and contextual reasoning by combining chain-of-thought prompting with dynamic tool calling. We provide a formal task definition in Section 3.1, followed by a comprehensive presentation of our specialized toolkit architecture designed for dynamic tool call in Section 3.2.

## 3.1 EGOCENTRIC LONG VIDEO REASONING TASKS

Compared to general exocentric videos, egocentric videos offer continuous, context-rich recordings from a first-person perspective, naturally documenting extensive temporal experiences including daily routines, social interactions, and object manipulations. This unique viewpoint requires sophisticated high-order inference to interpret actions, intentions, and contexts across substantial temporal spans, making it require reasoning models with strong temporal understanding and contextual integration capabilities. This necessitates a flexible reasoning framework that dynamically processes both visual information and contextual details through an intelligent tool-calling mechanism, determining which analytical approaches are most relevant for comprehending complex temporal narratives spanning multiple days of recorded experience.

In our task, we provide egocentric video spanning several days alongside questions posed at a specific query time. The system analyzes all preceding video content to generate accurate responses, simulating human temporal reasoning in real-life scenarios. This tool-based approach enables multimodal reasoning by leveraging contextual information across extended periods, requiring the system to choose optimal tools during the thinking process to effectively integrate perception, memory, and action when generating responses based solely on previously observed content.

## 3.2 DYNAMIC TOOL-CALLING

Current MLLMs struggle with extended egocentric content due to limited context windows, inadequate temporal understanding, and insufficient structured reasoning capabilities, preventing effective analysis of long-duration egocentric videos containing sparse events that require multi-step, context-aware interpretation. To address the inherent difficulty posed by the overly long context of long-form egocentric video reasoning, we adopt a *dynamic tool-calling* framework that empowers the LLM rather than an MLLM to invoke specialized perception tools on demand. Our approach enables the LLM to actively decompose complex queries, selectively retrieve relevant segments, and iteratively perform stepwise reasoning grounded in video observations. This modular design overcomes the context-length bottleneck of MLLMs while enabling the fine-grained, multi-turn reasoning essential for practical egocentric video understanding. Our framework leverages three complementary tools - one text-based and two visual-based - each addressing distinct temporal and perceptual dimensions of egocentric understanding. The text-based hierarchical RAG system handles longer temporal information retrieval, while the visual-based tools (Video-LLM and VLM) perform detailed visual analysis at different visual granularities.

`h-rag` : Our hierarchical system efficiently localizes relevant temporal information from the memory bank. Videos are first segmented into 30-second clips, each summarized via a video captioning model and temporally aligned with the ASR results as clip logs. These clip logs are hierarchically aggregated through a *bottom-up* generation process into multi-level granularity, creating comprehensive temporal summaries. The hierarchical structure facilitates effective *top-down* inference to locate and retrieve logs of relevant video segments, thus reducing computational load while preserving accuracy and temporal coherence across long egocentric videos spanning days. The system accepts specific search parameters, including temporal granularity, keywords, and time ranges for retrieval, returning the most relevant observations that match the query constraints.

`video-llm` : Our `video-llm` is a short-horizon visual-perception module that operates on local temporal windows ranging from a few seconds up to ten minutes. We sample each clip within the proposed time range at 1 FPS, keeping the input size compatible with modern multimodal language models and thus maintaining broad architectural flexibility. Given a question and its corresponding video segment, the tool correlates visual content with temporal context to produce detailed observations that capture dynamic interactions and sequential events, and, when possible, directly answers the query for the specified time range.

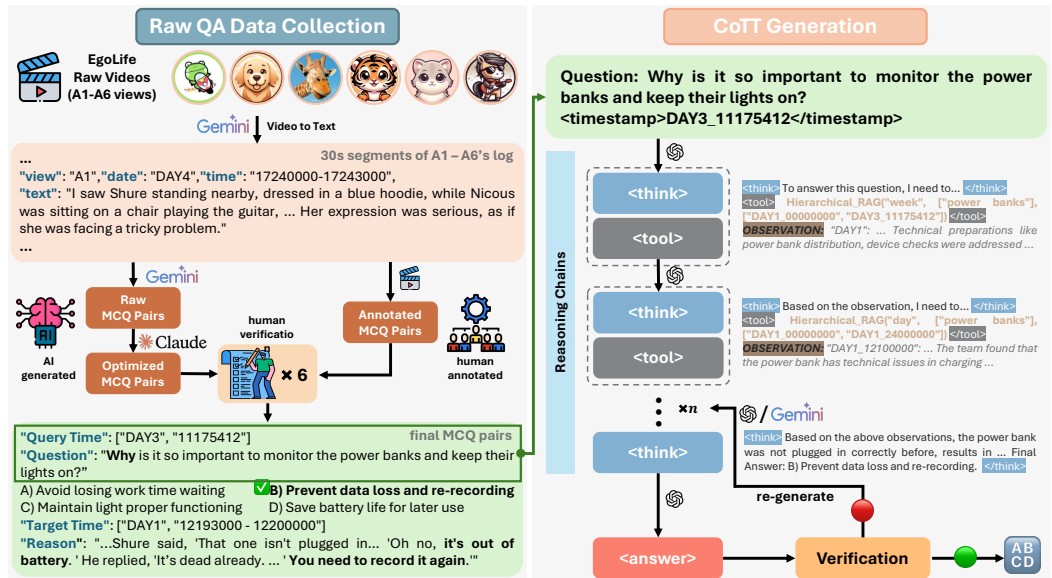

Figure 2: **Data generation pipeline of the Ego-R1 Data.** We first obtained raw QA pairs from both AI-generated and human-annotated sources based on 6 raw videos collected from 6 participants and the corresponding log. The verified and processed Multiple Choice Questions (MCQs) serve as the foundation of the Ego-R1 Data (left). We take questions without answers for Chain-of-Tool-Thought (CoTT) generation, which involves creating reasoning chains that include explicit thinking steps and dynamic tool-calling sequences (right).

`vlm` 📷: This general-purpose `vlm` operates at the finest temporal granularity, analyzing individual frames to extract high-resolution details like text on packaging, object attributes or specific visual elements missed in broader video analysis. It augments the temporal reasoning of `video-llm` with precise visual evidence for comprehensive egocentric understanding.

# 4 EGO-R1 DATA: CHAIN-OF-TOOL-THOUGHT (COTT) FOR VIDEO REASONING

To unleash the reasoning capabilities of LLM under the CoT prompting paradigm and to enable dynamic tool selection conditioned on current observations and past actions, we introduce Ego-R1 Data, a dataset designed to enable agentic tool-use with Chain-of-Tool-Thought (CoTT) reasoning chains. Figure 2 illustrates the data generation pipeline of the Ego-R1 Data, including raw QA data collection and CoTT generation. In this section, we define the structure of CoTT in Section 4.1, and provide details of Ego-R1 Data generation in Section 4.2.

## 4.1 CHAIN-OF-TOOL-THOUGHT (COTT)

Our goal is to generate synthetic CoTT data and use it to train multi-turn tool-use language models. We define a CoTT data $C$ as a sequence of steps $S_i$, where each step consists of a thought $T_i^{\text{th}}$, a tool $T_i^{\text{to}}$, and an observation $o_i$. A CoTT trajectory is defined as follows:

$$C = (S^0, S^1, \ldots, S^n), \quad S^i = \left( T_i^{\text{th}}, T_i^{\text{to}}, o_i \right) \tag{1}$$

where $C$ is a sequence of $n$ reasoning steps. At each step $i$, the agent will generate a thought $T_i^{\text{th}}$ and a tool call $T_i^{\text{to}}$ based on all the previous steps' observations $\{o_0, o_1, \ldots, o_{i-1}\}$ and the query $q$.

To formalize this reasoning process, we define two essential components that characterize how the agent operates: the action space, which specifies the available tools the agent can utilize, and the observation space, which captures the structured outputs returned from tool executions.

**Action space.** We define the action space $\mathcal{A} = F_j$ as a union of available tools to be used during reasoning. We use the three fundamental tools defined in Section 3.2: 1) `h-rag` for text-based long-range temporal retrieval, 2) `video-llm` for short-range video understanding, and 3) `vlm` for framewise image understanding, plus an auxiliary `terminate` tool for data generation only.

The `h-rag` tool retrieves relevant information from the current-view knowledge base by querying specified keywords within a target time window. By projecting long videos into a semantically and temporally structured language space, it rapidly pinpoints the approximate temporal interval of an event while summarizing sparse visual cues into a concise textual summary. The `video-llm` tool analyses short video segments specified by a query and an associated time window, providing detailed interpretations of local visual–temporal content. The `vlm` tool performs image-level analysis on a single frame selected by timestamp and query, providing precise, frame-specific visual details.

**Observation space.** At each reasoning step $i$, the agent receives an observation $o_i = \left(o_i^{\text{rag}}, o_i^{\text{vid}}, o_i^{\text{vlm}}\right) \in \mathcal{O}$, where each component $o_i^{\text{rag}}, o_i^{\text{vid}}, o_i^{\text{vlm}}$ represents the output of corresponding tool `rag`, `video-llm`, and `vlm`. The observation space $O = \{O^0, O^1, ..., O^n\}$ encompasses the collection of all tool outputs. Each tool call executes via the parsed arguments, producing observations that guide subsequent reasoning steps.

### 4.2 Data Generation

We carefully curate Ego-R1 Data, comprising 4.4K annotated question-answer pairs sourced from over 500 hours of egocentric videos recorded across six distinct first-person perspectives. We select 2.9K high-quality questions for CoTT generation. For each selected QA pair, we construct a CoTT trace that decomposes the reasoning process into interpretable steps, yielding an average of 7.42 tool calls per task. In total, 25K CoTT traces are generated, and subsequently used during the SFT stage to train our multi-turn tool-use language model.

**Ego-QA-4.4K.** Long-form egocentric videos are hard to collect in nature. Following the dataset construction pipeline of EgoLifeQA (Yang et al., 2025a), we collected 2.9K high-quality human-annotated data from 6 videos with distinct viewpoints. To expand the dataset scale, we employ proprietary models to analyze Automatic Speech Recognition (ASR) transcripts with video captioning outputs from the 30-second segments. These textual logs were combined and examined across various temporal granularities, spanning single or multiple days, to generate candidate questions with answers. Human annotators subsequently selected and cross-validated those QA pairs using Fleiss' kappa (Fleiss & Cohen, 1973), refining each query and its ground-truth answer according to a unified criteria of rationale coherence, importance, relevance, and difficulty level. In total, Ego-R1 Data comprises 4.4K question-answer pairs from both human-labeled and synthetic data sources.

**Ego-CoTT-25K.** We develop a systematic CoTT generation system to automatically generate CoTT data based on the selected question-answer pairs. By leveraging proprietary LLMs with longer context windows and stronger instruction-following capabilities, we enable the automatic generation of comprehensive reasoning chains that would otherwise be challenging to produce manually. In the CoTT generation system, each tool is exposed to the model as an executable function whose signature and semantics are implicitly embedded in the system. This design, paired with a textual system prompt (Table 13), prevents parsing errors during execution. The prompt also encodes the current viewpoint identity and enumerates the available tools. Given an input question $q$, the model iteratively generates reasoning steps $S_i = (T_i^{th}, T_i^{to})$, where $T_i^{th}$ denotes the thought and $T_i^{to}$ denotes the corresponding tool call with fully specified arguments (e.g., time ranges, keywords, sub-questions). All the proposed arguments are validated by a pre-verification module to ensure syntactic correctness. Once a call is emitted, its name and arguments are extracted via special tokens and dispatched to an external server for execution. The returned observation is then fed back to the model, guiding the next step and enabling dynamic, multi-turn tool use for egocentric long-video reasoning.

## 5 Ego-R1 Agent: Towards Tools Integrated Video Understanding Agent

Our goal is to train a language model capable of performing long-form video reasoning via a structured long-chain reasoning schema that automatically invokes multi-turn tool calls to collaboratively solve the problem. Inspired by the recent post-training techniques (Chu et al., 2025), we design our training framework with a two-stage strategy, with an illustration in Fig. 3.

### 5.1 Stage 1: Supervised fine-tuning (SFT)

In the first stage, we perform SFT on a pretrained language model using the synthetic CoTT dataset. This "cold-start" initialization equips the model with the foundational ability to produce correctly

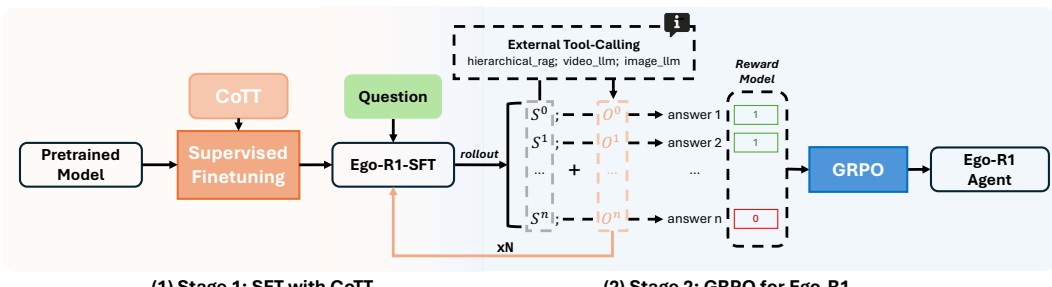

**(1) Stage 1: SFT with CoTT**     **(2) Stage 2: GRPO for Ego-R1**

Figure 3: **Overview of the two-stage training strategies in Ego-R1.** Ego-R1 employs a two-stage training approach: Stage 1 utilizes supervised fine-tuning with CoTT data to establish structured tool-calling capabilities, while Stage 2 applies multi-turn reinforcement learning with rule-based rewards to optimize iterative reasoning and tool execution across diverse question types.

formatted tool calls as prescribed by the CoTT reasoning schema. The CoTT data, presented in a structured, multi-turn conversational format, simulates realistic stepwise tool interactions, explicitly combining natural language reasoning with structured tool invocation. Each step in the reasoning trajectory consists of a thought enclosed within the special token `<think>...</think>`, followed by either a proposed tool call, enclosed within `<tool>...</tool>`, or an answer, enclosed with in `<answer>...</answer>`. The tool call is automatically parsed and executed by an external environment, which then returns an observation. This observation is formatted and fed back into the model as part of the input for the next reasoning step. After fine-tuning, the resulting *Ego-R1-SFT* model reliably produces well-formed tool calls and coherent step-by-step reasoning, laying the groundwork for subsequent reinforcement learning stage.

## 5.2 Stage 2: Reinforcement learning (RL)

To further improve the multi-turn tool-calling capabilities of our fine-tuned *Ego-R1-SFT* model, we adopt *Group Relative Policy Optimization* (GRPO) (Shao et al., 2024) to train the model. GRPO optimizes the model to maximize the expected final task reward while regularizing the variance of policy gradients across reasoning steps to encourage stable and coherent decision-making. Specifically, we define the GRPO objective as follows:

$$\mathcal{J}_{\text{GRPO}}(\theta) = \mathbb{E}_{[q \sim P(Q), \{o_i\}_{i=1}^G \sim \pi_{\theta_{\text{old}}}(O|q)]} \left[ \frac{1}{G} \sum_{i=1}^G \sum_{y=1}^T \frac{1}{|S_i^y|} \sum_{t=1}^{|S_i^y|} \left\{ \min \left[ \frac{\pi_\theta(S_{i,t}|q, I_y, S_{i,<t})}{\pi_{\theta_{\text{old}}}(S_{i,t}|q, I_y, S_{i,<t})} \hat{A}_{i,t}^y, \right. \right. \right.$$

$$\left. \left. \left. \text{clip}\left( \frac{\pi_\theta(S_{i,t}|q, I_y, S_{i,<t})}{\pi_{\theta_{\text{old}}}(S_{i,t}|q, I_y, S_{i,<t})}, 1-\varepsilon, 1+\varepsilon \right) \hat{A}_{i,t}^y - \beta \mathbb{D}_{\text{KL}}[\pi_\theta \| \pi_0] \right] \right\} \right]$$

In this equation, $\pi_\theta$ represents the policy model that generates reasoning tokens $S_i^y$ sequentially at turn $y$, where $i$ denotes the token position. The generation is conditioned on the preceding sequence $S^y * < i$, the observation $I_y$ at turn $y$, and the question $q$. The final reward $R * \text{final}(C, q)$ evaluates the correctness of the answer at the end of the reasoning chain $C$. The reference policy $\pi_0$ denotes the original model, and the KL divergence term $\text{KL}(\pi_\theta | \pi_0)$ regularizes the policy to prevent excessive drift from the initial parameters. The advantage estimates $\hat{A}_{i,t}^y$ are computed by standardizing rewards within each group $G$, subtracting the group mean and dividing by the group standard deviation.

During training, we generate rollout trajectories by sequentially executing tools based on the model's reasoning outputs, providing realistic stepwise observations that inform subsequent reasoning steps. Each rollout terminates when either a valid final answer is produced or the maximum step limit $N$ is reached. This training procedure enables the model to effectively generalize multi-turn tool usage, reflecting the iterative nature of egocentric long-video reasoning tasks. The resulting model after second-stage reinforcement learning training constitutes our final system, termed the *Ego-R1 Agent*.

Table 1: **Quantitative results on video question-answering benchmarks.** The proposed Ego-R1 model demonstrates superior performance across multiple metrics. Bold indicates best performance, underscored values show second best. The results from the 72B version of the model or using less frames are marked in gray. As some of the QA pairs in EgoLifeQA were used for CoTT generation and training, we excluded these from evaluation and retained only a clean subset for fair testing.

| Method | Size | Frames | Exocentric | Egocentric | | |
|---|---|---|---|---|---|---|
| | | | VideoMME (long) | EgoSchema | EgoLifeQA | Ego-R1 Bench |
| **Average durations** | | | 41 min | 3 min | 44.3 h | 44.3 h |
| *MLLMs* | | | | | | |
| LongVA (Zhang et al., 2024a) | 7B | 64 | 45.0 | 44.1 | 33.0 | 23.0 |
| LLaVA-Video (Zhang et al., 2024b) | 7B | 64 | 61.5 | 57.3 | 36.4 | 29.0 |
| LLaVA-OneVision (Li et al., 2024a) | 7B | 1 FPS | 60.0 | 60.1 | 30.8 | 31.6 |
| InternVideo2.5 (Wang et al., 2025b) | 8B | 512 | 53.4 | 63.9 | 33.0 | 34.0 |
| Gemini-1.5-Pro (Team et al., 2024) | - | - | **67.4** | **72.2** | **36.9** | 38.3 |
| *RAG Methods* | | | | | | |
| LLaVA-Video + Video-RAG (Luo et al., 2024) | 7B | 64 | 46.0 | 66.7 | 30.0 | 29.3 |
| LongVA + Video-RAG (Luo et al., 2024) | 7B | 64 | 55.7 | 41.0 | 26.0 | 31.0 |
| *Reasoning Models* | | | | | | |
| Video-R1 (Feng et al., 2025b) | 7B | 64 | 50.8 | - | 34.0 | 20.0 |
| *Video Agents* | | | | | | |
| VideoAgent (Wang et al., 2024a) | - | 8 | 50.8 | 54.1 | 29.2 | 32.6 |
| LLaVA-OneVision + $T^*$ (Ye et al., 2025) | 7B | 8 | 46.3 | 66.6 | 35.4 | 35.6 |
| *Ours* | | | | | | |
| **Ego-R1** | **3B** | - | 64.9 | 68.2 | 36.0* | **46.0** |

## 6 EXPERIMENTS

### 6.1 EXPERIMENT SETUP

To evaluate the effectiveness of the CoTT reasoning traces in answering the ultra-long video understanding question, we utilize Qwen-2.5-3B-Instruct as our base model. To mitigate the hallucination problem caused by the increasing CoTT length, we introduce an additional summary model with a longer context window length to help conclude the reasoning trace to answer the question.

**Benchmarks.** We evaluate the performance of Ego-R1 Agent on three existing long video understanding benchmarks covering both exocentric and egocentric views: Video-MME (long w/o subtitle) (Fu et al., 2024), EgoSchema (Mangalam et al., 2023), EgoLifeQA (Yang et al., 2025a). Among them, Vide-MME has a third-person view, and the rest have the first-person view. We follow the same paradigm as h-rag to generate the knowledge base for each video in these benchmarks. The hierarchy depth of each memory bank varies by datasets: only EgoLifeQA contains videos long enough to necessitate day-level summaries, while others extend to 10-minute-level or hour-level summaries at most. To further evaluate the capability of Ego-R1 Agent in handling multi-perspective and long temporal reasoning question answering tasks, we establish **Ego-R1 Bench**, a reasoning based benchmark for ultra-long egocentric video understanding. Distinct from Ego-R1 Data, Ego-R1 Bench comprises 300 QAs evenly distributed across six first-person perspectives. For each perspective, Ego-R1 Bench includes a balanced mixture of human-labeled and human verified QAs.

**Comparison Methods.** We benchmark Ego-R1 Agent against recent representative approaches, including MLLM-based video understanding methods (Zhang et al., 2024a;b; Li et al., 2024a; Wang et al., 2025b; Team et al., 2024), RAG-based method (Luo et al., 2024), reasoning model (Feng et al., 2025b) and video agents (Wang et al., 2024a; Ye et al., 2025). For each question, we restrict the input to video content occurring *before* the query timestamp, ensuring causal consistency in all comparisons. To ensure fair comparison across methods with different architectural constraints, we adopt an adaptive frame-sampling protocol: 1) Standard frame-based MLLMs (Zhang et al., 2024b;a; Feng et al., 2025b) and LLaVA-OneVision (Li et al., 2024a) receive 64 uniformly sampled frames per query; 2) Video-RAG (Luo et al., 2024) uses its native setting of 64 frames; 3) Higher-capacity models such as InternVideo2.5 (Wang et al., 2025b) and Gemini 1.5 Pro (Team et al., 2024) are provided with 512 uniformly sampled frames; 4) Agent-based methods that rely on caption-guided key-frame selection (Wang et al., 2024a; Ye et al., 2025) are supplied with 1 024 uniformly sampled frames, recomposed into 1 FPS videos. This protocol equalizes input budgets while respecting each model's architectural constraints.

## 6.2 RESULTS

Table 1 presents a quantitative comparison of Ego-R1 with state-of-the-art video understanding models on both exocentric and egocentric benchmarks. Ego-R1 achieves the best or second-best score on three of the four datasets, despite using far fewer parameters than most competitors.

**Exocentric setting.** On VideoMME (long), whose clips average 41 min, Ego-R1 achieves 64.9% accuracy, which is the highest score among open-weight models and second overall, falling behind only the proprietary Gemini-1.5-Pro (67.4%). It surpasses other public MLLMs, such as LLaVA-Video (61.5%) and InternVideo2.5 (53.4%), while using less than half their parameter count. These results indicate that, although Ego-R1 is trained in an egocentric regime, it generalizes effectively to exocentric settings.

**Egocentric settings.** Ego-R1 achieves the highest accuracy on the proposed egocentric long video reasoning benchmark - Ego-R1 Bench (with an average time of 44.3 h), achieves 46.0% accuracy. This result exceeds Gemini-1.5-Pro by 7.7% and surpasses the strongest open baseline, LLaVA-Video, by 17.0%, underscoring the benefit of hierarchical retrieval and multi-turn tool calling for reasoning tasks with sparsely distributed events. On EgoSchema (3 min clips), Ego-R1 records 68.2%, second only to Gemini (72.2%); on EgoLifeQA we obtain 36.0% after removing any training overlap, comparable with LLaVA-Video (36.4%) and approaching Gemini (36.9%).

**Analysis.** Both frame-based MLLMs and RAG variants exhibit marked performance drops on Ego-R1 Bench, and agent-based approaches remain in the 32 - 36% range, well below the 46% achieved by Ego-R1. These findings indicate that agent-based approaches provide a more effective solution for long-video reasoning tasks, while our CoTT style dynamic tool calling, enables even a compact 3B model to conduct reliable, long-horizon reasoning over hours-long egocentric video.

## 6.3 ABLATION STUDY

To better understand the contribution of different training components in Ego-R1, we conduct ablation studies using identical base models under varying training regimes. Specifically, we compare models trained with: (1) SFT only, (2) RL only, and (3) a combination of both. Quantitative results are reported in Table 2.

Table 2: **Ablation study on different training regimes.** We use Qwen-2.5-3B-Instruct as our base model.

| Base Model | Training Regimes | | Acc.% | Format Acc.% |
|---|---|---|---|---|
| | SFT | RL | | |
| | – | – | 1.4 | 4.3 |
| Qwen-2.5 | – | ✓ | 0.0 (↓1.4) | 13.3 (↑9.0) |
| 3B-Instruct | ✓ | – | 34.3 (↑32.9) | 100.0 (↑95.7) |
| | ✓ | ✓ | 46.0 (↑44.6) | 100.0 (↑95.7) |

The zero-shot base model achieves only 1.4% task accuracy and 4.3% tool-call format accuracy. Vanilla RL training without CoTT supervision surprisingly drops task accuracy to 0% while improving format accuracy by 9%, showing the model learns tool-call structure but produces ungrounded predictions without reasoning supervision. Conversely, SFT with CoTT data significantly improves both metrics even with limited training (3 epochs), demonstrating that structured reasoning demonstrations are essential for establishing multi-step reasoning foundations. Complete ablation results regarding tools and efficiency analysis are detailed in the supplementary materials.

## 7 CONCLUSION AND OUTLOOK

We introduce Ego-R1, a novel framework addresses challenges in long-horizon egocentric video reasoning through its Chain-of-Tool-Thought approach, which demonstrates that decomposing complex reasoning into modular, tool-grounded steps creates a more robust foundation for video understanding than traditional methods. This integration of structured reasoning with dynamic tool-calling not only enhances model interpretability but also reveals promising directions for future multimodal AI systems. The superior performance across temporal coverage and information retention suggests that hybrid architectures combining symbolic and neural components may be essential for tackling open-world, ultra-long video understanding. Beyond immediate applications in egocentric video analysis, Ego-R1 points toward broader implications for human-AI collaborative systems where transparent reasoning processes are critical, boosting the potential for life-oriented assistants that accompany humans over very long timeframes, making them genuinely useful and seamlessly integrated into everyday life.

## ETHICS STATEMENT

This work complies with the ICLR Code of Ethics. Our research primarily focuses on dataset and methodological contributions to multimodal long-video reasoning, evaluation, and tool orchestration. We do not foresee direct negative societal impact, but acknowledge the following considerations:

- **Data Sources:** Our dataset is build on publicly available dataset, Egolife, and our experiments are conducted on publicly available benchmarks (e.g., VideoMME, EgoSchema, EgoLifeQA). We ensure that all datasets used are subject to their respective licenses and terms of use. EgoLifeQA annotations were created by trained annotators following internal guidelines, with care to avoid sensitive or personally identifiable information.

- **Privacy and Anonymity:** Egocentric video datasets may contain personal environments or daily activities. We relied only on data that had already undergone institutional collection protocols and release agreements. No additional personally identifiable information was introduced.

- **Bias and Fairness:** As with all large-scale vision-language data, inherent biases may exist (e.g., cultural, demographic, or activity distribution biases). We note these limitations and encourage future work to study fairness and inclusivity in egocentric video understanding.

Overall, this work aims to advance the scientific understanding of long-video multimodal reasoning while being mindful of privacy, fairness, and responsible use.

## REPRODUCIBILITY STATEMENT

We acknowledge the reproducibility of our work. Our framework for long-horizon egocentric video reasoning relies on large-scale datasets, some of which are publicly available while others are still under release preparation. To facilitate replication, we will release the full training and orchestration codebase with clear documentation. Key hyperparameters and evaluation settings are reported in the main paper, and additional ablation results are included in the appendix. Our findings remain verifiable in principle, and we encourage future research to build upon and validate them as resources and artifacts are released.

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

APPENDIX

The supplementary document provides (1) details of the hierarchical RAG during the dynamic tool-calling in Section B; (2) additional data statistics analysis and comparison in Section C; (3) comprehensive implementation details including the prompts we used for data generation and training, experiment setup in Section D; (4) additional experiments and ablation studies in Section E; (5) a comparison among our framework and others F(6) future works G, respectively.

## A   USE OF LARGE LANGUAGE MODELS

During dataset curation for Ego-CoTT-25K, LLMs were used for synthetic CoTT trjectory generation and cross-validation with quality checking, and more detailed can be found in Section D.1. In addition, we utilized GPT-5 to polish the manuscript for clarity and readability. For example, we applied the following prompt to refine the initial draft: *"Please check the grammar of the following draft and revise if necessary."*

## B   HIERARCHICAL RAG

As mentioned in the paper, during the dynamic tool-calling, our framework leverages three complementary tools - one text-based and two visual-based - each addressing distinct temporal and perceptual dimensions of egocentric understanding. In this section, we provide additional details.

To facilitate more efficient and accurate reasoning over extremely long videos, we construct a hierarchical RAG system, as shown in Figure 4. Specifically, for each video $V$, we first segment it into 30-second clips $v_i$, while preserving natural recording boundaries. The memory bank of the RAG system is built upon these 30-second clips. For each clip $v_i$, we employ a VLM (Gemini 1.5 Pro (Team et al., 2024)) to generate comprehensive summaries $S_{\text{clip},i}$ that include both dense captions of visual content and transcripts of spoken dialogue. We then leverage an LLM (GPT 4 (Achiam et al., 2023)) to progressively synthesize these fine-grained summaries into increasingly *coarser* temporal resolutions, while respecting natural recording boundaries at each level. Typically, the clip-level summaries are combined into 10-minute summaries $S_{10\text{min},m}$, which are further aggregated into hourly summaries summaries $S_{\text{hour},h}$, and finally sorted into day-level summaries $S_{\text{day},d}$.

The hierarchical RAG system serves as a critical component within our CoTT framework for long video reasoning. During the reasoning process, when a <think> step determines that the RAG system is the optimal tool to retrieve the related information, it formulates a query $q = $ (level, [keywords], starting_timestamp, query_timestamp). Here, level designates the temporal granularity, with 'week' targeting a specific day within the week, 'day' targeting a specific hour range within a day, and 'hour' targeting a specific 10-minute segment within an hour. The keywords specify the search terms, while timestamps are represented with precise DAY_X specification and HHMMSSss format. The subsequent <tool> step passes this query to the RAG system, initiating the hierarchical retrieval process. The retrieval follows a top-down approach, cascading from the specified entry level through the hierarchy and returning relevant summaries. Such hierarchical navigation aligns naturally with the temporal structure of extremely long egocentric videos, which inherently follow daily patterns of human activities. Keywords retrieval at each stage is conducted through LLM-based (GPT4) keyword matching on the textual summaries, with time-indexed metadata maintained throughout the hierarchy to enable fast localization. Importantly, the <think> step determines both the initial level for the search and whether to continue to *finer* levels based on the summaries returned. When sufficient information is found at a *coarser* level (day or hour), the <think> step may choose not to proceed to *finer* granularities. After receiving summaries from the RAG system, the subsequent <think> step evaluates these results and determines whether to extract answers directly or employ additional tools for further analysis and evidence localization. This hierarchical approach significantly reduces computational overhead by avoiding exhaustive search across the entire video corpus and by terminating the search at the earliest level that yields sufficient information, while maintaining high keywords retrieval accuracy through the preservation of temporal relationships.

Figure 4: **Overview of the Hierarchical RAG system.** Based on the raw video and its 30-second clips, we generate the memory bank for each video from its 30-second-level summaries to day-level summaries. During the keywords retrieval, the system searches efficiently by starting with day-level summaries and drilling down to 10-minute segments as needed.

## C  DATA STATISTICS

We categorise the benchmark difficulties based on the temporal gap between the *query time* (when a question is raised) and the *target time* (where the corresponding answer can be located in the video). This definition captures how long a model must effectively "look back" to resolve the query. Intuitively, shorter spans require only local memory, while longer spans test long-term temporal reasoning and retrieval.

Table 3 presents both the distribution of difficulty levels (left) and summary statistics of the temporal spans (right). We observe that around one-third of the questions fall in the **Hard** (1–3 days) category, confirming that the benchmark deliberately stresses beyond short-term context. Another notable portion (29%) are **Easy** queries within a 1–6 hour range, which reflects realistic daily routines where events are correlated over shorter windows. Meanwhile, **Very Hard** cases (12.7%) extend beyond 3 days, forcing the agent to integrate scattered evidence across multi-day activities.

Table 3: **Question Difficulty Distribution (Left) and Key Time-span Statistics (Right).**

| Difficulty | Time Span | Count | Percentage |
|---|---|---|---|
| Very Easy | < 1 hour | 25 | 8.3% |
| Easy | 1–6 hours | 86 | 28.7% |
| Medium | 6–24 hours | 62 | 20.7% |
| Hard | 1–3 days | 89 | 29.7% |
| Very Hard | > 3 days | 38 | 12.7% |

| Statistic | Value |
|---|---|
| Mean time span | 28.74 hours ( 1.2 days) |
| Median time span | 17.83 hours ( 0.74 days) |
| Range | 0.03–142.20 hours (2 min–5.9 days) |

The key statistics further illustrate this long-horizon challenge. The mean span is 28.7 hours (~1.2 days), significantly larger than typical video QA datasets that often operate within a single clip or short session. The median span of 17.8 hours indicates a heavy skew toward longer dependencies, while the range stretches from as short as 2 minutes to nearly 6 days. These statistics highlight that Ego-R1 must adaptively combine local perception with long-term retrieval, mirroring the real-world demands of egocentric reasoning.

## D  IMPLEMENTATION DETAILS

### D.1  ENVIRONMENT SETUP

During the dataset construction phase, for CoTT data generation and keywords retrieval in our hierarchical RAG system, we utilize GPT-4.1 (OpenAI, 2025a) as the routing LLM. During raw QA data collection for Ego-R1 Data and Ego-R1 Bench, we use Gemini 1.5 Pro for video-to-text processing, Gemini 2.5 Pro (Google DeepMind, 2025) for raw MCQ pairs generation, and Claude3.5 Sonnet (Anthropic) for MCQ pairs post-processing. In the main experiment, we use

LLaVA-Video (Zhang et al., 2024b) during training as the backbone of Video-LLM, and Qwen2.5-VL-7B-Instruct (Bai et al., 2025) as the backbone of VLM. During the inference phase, we use Gemini 1.5 Pro (Team et al., 2024) as the backbone of Video-LLM, and GPT-4o (OpenAI, 2024) as the backbone of VLM. We conducted the training of SFT and RL on 4 NVIDIA 80GB A100 GPUs, and experiments for baseline comparison on 1 NVIDIA 80GB A100 GPU.

### D.2 DATA GENERATION

We leverage a proprietary model GPT-4.1, with the AutoGen framework (Wu et al., 2023) to systematically generate the CoTT data. The AutoGen framework supports tool-use without parse or execution failures by leveraging structured message passing and function calling via standardized protocols like OpenAI function-calling or JSON schema-based interfaces. Unlike prompt-only approaches that rely on natural language parsing, AutoGen agents interact with tools through well-defined wrapped functions, ensuring that function arguments are syntactically and semantically valid before execution. The framework includes built-in validation, exception handling, and modular components (e.g., user proxy, assistant agent, and tool agent) that collaboratively manage errors, retries, and fallbacks. This design reduces the likelihood of malformed calls and enhances robustness in executing complex multi-step tool-use workflows. We have attached the prompts used for data generation in Table 13.

Our annotation pipeline has combined several quality control methods. Our corpus comprises (1) QA items and (2) CoTT (Chain-of-Tool-Thought) traces.

1. QA annotation (primary quality target: answer correctness).

- **Unified guidelines.** Annotators follow a single rubric covering: factual correctness, time-span validation, non-triviality/ambiguity checks, and exclusion of questions with insufficient evidence.

- **Agreement.** We are using Fleiss's kappa 4 to ensure data quality consistency across annotators.

2. CoTT traces (primary focus: format + faithfulness).

- **Automated trace engine.** We contribute a fully-automated CoTT data engine that can generate CoTT traces from labeled & refined QA pairs. The format correctness check is encoded as a rule-based function to ensure each tool is invoked and executed correctly.

- **Postprocess (Quality filters).** Additional rule-based filters remove traces with system errors, incorrect final answers, or over-long/looping reasoning paths. Therefore, we obtain the final Ego-CoTT-25K.

In consideration of the privacy issues and double-blind policy, we will release the inter-annotator agreement, annotation guidelines, raw agreement metrics, and error taxonomy upon paper acceptance.

### D.3 RAG CONSTRUCTION FOR OTHER BENCHMARKS

One key component of our framework is the personalized RAG system tailored to each egocentric perspective, which requires additional effort to build and maintain. To ensure fair comparisons across benchmarks, we adapt the RAG construction strategy to the temporal characteristics of each dataset. Since the average timespan per question varies across benchmarks, we adjust the indexing granularity accordingly that proportional to the average length of the benchmark.

For the EgoLife (Yang et al., 2025a) benchmark, where the recording time span and task setting align with our framework with average time span for 44.3 hours, we adopt a hierarchical RAG structure with consistent temporal levels: *week → day → hour → 10-minute*. This allows for flexible, top-down retrieval over long egocentric sequences while preserving temporal precision. For other benchmarks with shorter video durations or coarser question alignment, we use simplified RAG levels to match the dataset's temporal scope.

For the VideoMME (Fu et al., 2024) benchmark, we tested on the long video subset, which has an average time span of 41 minutes. We divide videos into 30-second segments, and use Gemini-1.5-Pro to summarize each segment by combining visual content and transcribed dialogue. We then use GPT-4.1 to hierarchically aggregate these segments into 10-minute summaries. The hierarchical RAG's temporal structure would be: *10-minute → 30-second*. This two-level summarization enables our RAG system to be adapted to a shorter task duration while maintaining hierarchy.

For EgoSchema (Mangalam et al., 2023) benchmark, we segment the videos into 30-second clips and summarize each using LLaVA-video-7b (Zhang et al., 2024b), operating on 1 FPS sampled frames. The summaries are written in the first person to reflect egocentric perspective and are used as memory entries in our RAG system. Given the limited timespan of questions in this dataset, no additional hierarchical summarization is applied beyond the clip level.

# E  ADDITIONAL EXPERIMENTS

## E.1  ABLATION STUDIES

We conducted ablation studies to assess the contribution of different tools in Ego-R1. Table 4 summarizes two key aspects: (left) replacing LLaVA-Video with Gemini-1.5-Pro improves accuracy from 43.7% to 46.0%, showing that stronger video-LLM backbones enhance performance while the framework remains robust across choices; (right) using only RAG reduces accuracy to 39.7%, confirming that retrieval alone is insufficient and that visual grounding tools are essential for long-horizon reasoning.

Table 4: **Ablation studies on tool use.** (Left) Substituting Gemini-1.5-Pro for LLaVA-Video improves accuracy, demonstrating adaptability to stronger video-LLM backbones. (Right) Using only RAG lowers performance to 39.7%, while combining retrieval with visual tools restores accuracy to 46.0%, confirming the necessity of a multi-tool architecture for long-horizon reasoning.

| **Method** | Video_LLM | Ego-R1 Bench |
|---|---|---|
| **Ego-R1** | LLaVA-Video(Zhang et al., 2024b) | 43.7 |
| | Gemini-1.5-Pro(Team et al., 2024) | 46.0 |

| **Method** | Tool-used | Ego-R1 Bench |
|---|---|---|
| **Ego-R1** | RAG only | 39.7 |
| | Full | 46.0 |

Table 5: **Comparison with open-source tools.** Ego-R1-Agent using LLaVA-Video and Qwen-2.5-VL yields consistent improvements over the single-tool baseline.

| Agent | H-RAG | Video-LLM | VLM | Bench (%) |
|---|---|---|---|---|
| LLaVA-Video | – | – | – | 28.0 |
| Ego-R1 Agent 3B | ✓(Qwen2.5-14B) | – | – | 31.3 (↑3.3) |
| Ego-R1 Agent 3B | ✓(Qwen2.5-14B) | LLaVA-Video | Qwen-2.5-VL | 34.3 (↑6.3) |
| Ego-R1 Agent 3B | ✓(GPT-4) | LLaVA-Video | Qwen-2.5-VL | 43.7 (↑15.7) |

We further analyze open-source tool configurations in Table 5. LLaVA-Video alone achieves 28.0%. Adding hierarchical RAG with Qwen2.5-14B increases accuracy to 31.3%, and including both LLaVA-Video and Qwen-2.5-VL raises it to 34.3%, highlighting the complementary roles of video- and image-level grounding.

Substituting GPT-4 as the orchestrator yields 43.7%, demonstrating that stronger controllers substantially amplify tool effectiveness while preserving modularity.

Table 6: **Comparison with proprietary tools.** Gemini-1.5-Pro and GPT-4/4o provide strong baselines, while Ego-R1-Agent integration further boosts performance.

| Agent | H-RAG | Video-LLM | VLM | Bench (%) |
|---|---|---|---|---|
| Gemini-1.5-Pro | – | – | – | 38.3 |
| Ego-R1 Agent 3B | ✓(GPT-4) | – | – | 39.7 (↑1.4) |
| Ego-R1 Agent 3B | ✓(GPT-4) | LLaVA-Video | Qwen-2.5-VL | 43.7 (↑5.4) |
| Ego-R1 Agent 3B | ✓(GPT-4) | Gemini-1.5-Pro | GPT-4o | 46.0 (↑7.7) |

Table 6 examines proprietary settings. Gemini-1.5-Pro alone achieves 38.3%. Adding hierarchical RAG with GPT-4 improves performance slightly to 39.7%, while introducing open-source vision tools raises it further to 43.7%. The best result, 46.0%, is obtained when GPT-4 orchestrates Gemini-1.5-Pro and GPT-4o jointly. Taken together with the open-source results, these findings reinforce two consistent trends: (1) hierarchical RAG is a recurring driver of performance, and (2) the modular design of Ego-R1enables effective integration of both open-source and proprietary components, with stronger controllers systematically amplifying the gains from tool chaining.

## E.2  QUALITATIVE RESULTS

In the attached demo video, we show two examples of our step-by-step reasoning with agentic tool-use. Additionally, we present the reasoning comparison with Video-R1 (Feng et al., 2025b) in Figure 5. As illustrated in the visual demos, our method offers a more interpretable reasoning process through the explicit CoTT outputs. These structured reasoning chains reveal how collaborative tool

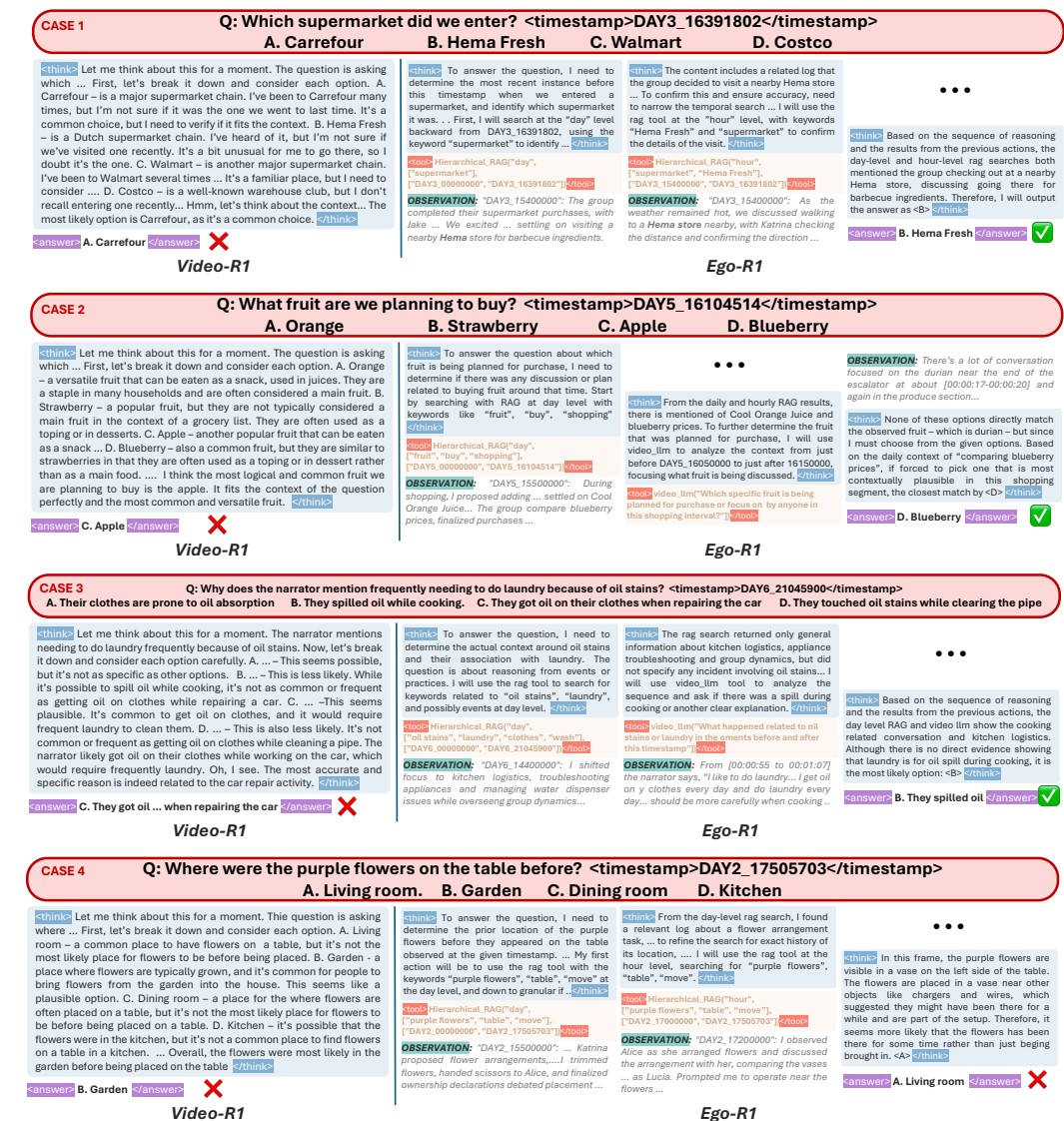

Figure 5: **Qualitative results comparison with Video-R1.** Case 1-3 illustrate successful examples where Ego-R1 outperforms Video-R1 by producing more detailed, interpretable step-by-step reasoning chains through dynamic tool-calling. In contrast, Case 4 highlights a failure case from Ego-R1 Agent. Although the observation in Step 1 correctly identified relevant information near timestamp `DAY2_15500000`, the subsequent tool call failed to adjust the temporal range accordingly, resulting in an incorrect or suboptimal retrieval in the next step, leading to the final error answer.

use contributes to improved performance on complex video reasoning tasks. A common failure case is shown in the last example in Figure 5. Although the observation from step 1 indicates the agent has successfully located the time range that possibly has relevant information, in the following step, the agent does not explore further from the observed time range.

E.3 EFFECTIVENESS OF THE PROPOSED METHOD

Our contributions in the proposed method are twofold: (1) we propose a method capable of handling ultra-long video inputs spanning days to weeks in a more reliable and explainable way, and (2) we achieve state-of-the-art or strong second-best performance on several long/ultra-long video benchmarks. Existing video-language models such as Gemini-1.5 Pro are fundamentally limited by token constraints, restricting their input to approximately one hour of video (Gemini-1.5 Pro

maximum input tokens: 1,048,576 (Team et al., 2024)). In contrast, the videos in EgolifeQA and our proposed Ego-R1 Bench average 44.3 hours, significantly exceeding these limits. As detailed in our evaluation setup in the experiments section, we use sampled frames to enable fair comparison, as full-length inputs remain infeasible for current models. The frames are sampled by this: $\text{Frames} = \left\{ F_k \ \middle| \ t_k = t_{\text{start}} + \frac{k}{K-1}(t_{\text{end}} - t_{\text{start}}), \ k = 0, 1, \ldots, K-1 \right\}$, where $t_{\text{start}}$ and $t_{\text{end}}$ are the time bounds returned by RAG; $K$ is the total number of frames you wish to sample; $F_k$ is the frame at timestamp $t_k$. However, even after downsampling, critical information required to answer the questions is often omitted, challenging the reliability and interpretability of the conventional models. Our method uses agentic tool-use to obtain a more accurate and reliable inference process to answer this novel and challenging task.

Importantly, our work is the first to address ultra-long egocentric video reasoning at this scale, representing a fundamentally different and more challenging task compared to existing benchmarks designed for shorter videos. Even with a lightweight 3B orchestrating model, Ego-R1 matches or exceeds state-of-the-art performance across the majority of benchmarks. In the supplementary material, we include an ablation study demonstrating the contributions of the tools, and we've integrated a summary module to terminate reasoning loops when the agent fails to stop on its own, further boosting the performance.

Table 7: **Efficiency comparison.** Unlike prior works that rely on a fixed number of sampled frames, Ego-R1 Agent dynamically invokes visual tools for targeted video segments, enabling efficient ultra-long video reasoning.

| Method | Avg. Frames | Time (s) | Device |
|---|---|---|---|
| LLaVA-Video | 64 | 1.16 | RTX A6000 |
| VideoRAG + LLaVA-Video | 64 | 22.03 | RTX A6000 |
| Video-R1 | 64 | 24.84 | NVIDIA A100 |
| $T^*$ + LLaVA-OneVision | 512 | 186.75 | RTX A6000 |
| Gemini-1.5-Pro | 1fps / max6k | 62.9 | NVIDIA A100 |
| Video Agent | 1,024 | 256.7 | NVIDIA A100 |
| Ego-R1 Agent 3B (open-source) | 48.7 | 82.4 | RTX A6000 |
| Ego-R1 Agent 3B (proprietary) | 411.3 | 66.2 | RTX A6000 |

As our method adopts a chain-of-thought (CoT)-style multi-step reasoning process, there is an inherent trade-off between inference efficiency and performance since it increases the output length and may impact latency. To address this, we have made the number of reasoning steps controllable via a hyperparameter, allowing users to balance computational cost and accuracy based on their specific use case. Our trained Ego-R1 Agent has 3B parameter size, which is more computation-efficient compared with other tool-use models. Regarding the time and resource efficiency, we conducted an efficiency comparison test on the baseline models and the proposed method. In addition, the number of frames used during the visual tools (video-llm & vlm) execution is dynamically adjusted according to the tool's effective time-duration window. We show the quantitative analysis of the frames used and inference time in Table 7. Despite being a compact 3B model, the proposed Ego-R1 Agent achieves superior performance with optimal frame efficiency: using only 48.66 frames on average with open-source tools (vs. 64 frames required by baseline methods) while significantly outperforming them. When using proprietary tools, our agent processes 411.3 frames, substantially fewer than methods requiring 512-1024 frames yet still achieves state-of-the-art results. The inference time reflects this efficiency-performance trade-off: while some methods achieve faster inference (<30s) with poor accuracy, our approach balances both aspects, delivering SOTA performance in 66.15s with proprietary tools.

### E.4 Tool-used Quantitative Results

We analyzed 300 questions and 3,002 total tool invocations to understand how Ego-R1 dynamically composes its pipeline. Regarding the tool-use patterns:

- **Tool Frequency:** RAG is the most frequently used tool (46.6%), followed by VLM (27.3%) and Video LLM (26.1%).

- **Tool Sequences:** Questions typically use about 10 tools on average, with complex reasoning chains often following patterns like "rag → video_llm → vlm".

- **Response Lengths:** Most answers are short (median 10 characters, likely multiple choice), but some are very long (up to 39,930 characters).

Regarding the tool frequency and adaptivity, we provide statistics in Table 9. We show that RAG underpins nearly half of all calls, confirming its role as the inference backbone together with the tool ablation study on the effectiveness. In addition, VideoLLM and VLM split the remaining workload, collaboratively refining temporal and visual details.

Table 8: **Tool usage statistics.**

| Metric | Value |
|--------|-------|
| Questions | 300 |
| Tool calls | 3,002 |
| Avg./Q | 10.0 (max 14) |

Regarding how search patterns correlate with performance, and how long-range tools (rag & video-llm) localization accuracy impacts the results, we provide statistics of top search patterns in Table 10.

We also show the localization ability of long-range tools (rag & video-llm) in Table 11. From these analyses, we can infer

Table 9: **Tool Frequency and Adaptivity.**

| Tool | # Calls | % Total | When Most Used |
|------|---------|---------|----------------|
| **RAG** | 1,399 | 46.6% | Context retrieval, fallback |
| **VLM** | 819 | 27.3% | Mid-chain visual grounding |
| **VideoLLM** | 784 | 26.1% | Deep video reasoning |

Adaptive pattern:
– Fact Qs: rag → rag → video-llm → vlm
– Visual Qs: video-llm invoked earlier

that video_llm is more localization-sensitive: the gap between "evidence contained" vs. "not contained" for video_llm is 11.5%, compared to just 4.7% for RAG. This shows that video_llm's accuracy depends heavily on precise temporal retrieval. Furthermore, RAG provides coarse, robust retrieval with a mean of ∼1,688 min and only a small accuracy swing (35.7% vs. 40.4%), reliably narrowing down the search window but not fine-grained enough for direct reasoning. It confirms our assumption that its role is to gather context rather than deliver the final answer.

Overall, we show that our agent can adapt its tool-chain based on question type and intermediate results, and reflects after each call, re-invoking tools when confidence or completeness falls below a threshold.

We summarize the tool frequency, response lengths and tool usage sequences in Table 12.

Table 10: **Top Search Patterns.**

| Pattern | Count | Accuracy (%) |
|---------|-------|--------------|
| rag → rag → video_llm → vlm | 49 | 30.6 |
| rag → rag → vlm | 11 | 27.3 |
| rag → video_llm → vlm | 11 | 27.3 |
| rag → rag → vlm → video_llm | 10 | 40.0 |
| rag → rag → video_llm → video_llm → vlm | 6 | 33.3 |
| rag → rag → video_llm → vlm → rag → rag → rag → video_llm | 5 | 20.0 |

Table 11: **Long-range Tools (rag & video-llm) Localization Accuracy.**

| Tool | Mean temporal distance(min) | Median temporal distance(min) | Acc.(evidence contained)(%) | Acc.(evidence not contained)(%) |
|------|------|------|------|------|
| rag | 1688.57 | 949.40 | 35.7 | 40.4 |
| video-llm | 1568.95 | 792.83 | 46.4 | 34.9 |

# F    COMPARISON WITH OTHER FRAMEWORKS

We compare Ego-R1 with existing long-video understanding and reasoning frameworks in Table **??**. Prior approaches such as LongVA (Zhang et al., 2024a) and LLaVA-Video (Zhang et al., 2024b) provide only short-video reasoning without mechanisms for retaining information or modeling temporal dependencies. More recent methods including $T^*$ (Ye et al., 2025), Video-RAG (Luo et al., 2024), and VideoAgent (Wang et al., 2024a) incorporate retrieval or memory components, but they remain limited in scalability to ultra-long inputs and lack interpretable reasoning chains. Video-R1 (Feng et al., 2025b) introduces explicit reasoning but does not address information retention or temporal grounding.

By contrast, Ego-R1 achieves all five requisites: it scales to ultra-long videos, preserves critical information through hierarchical RAG, provides interpretable reasoning via agentic tool calls, and maintains temporal awareness through localized retrieval. Moreover, its modular design allows seamless integration of both open-source and proprietary tools, ensuring adaptability across backbones

Table 12: **Summary of Tool-usage.**

```
## SUMMARY
Total Questions: 300
Total Tool Calls: 3002
Total Answers: 901
Total Thinking Sections: 2643

## TOOL FREQUENCY
rag: 1399 calls (46.6%)
vlm: 819 calls (27.3%)
video_llm: 784 calls (26.1%)

## RESPONSE LENGTHS
Answers:
- mean: 5882.7 characters
- median: 10.0 characters
- min: 1.0 characters
- max: 39930.0 characters
Thinking Sections:
- mean: 1011.0 characters
- median: 1078.0 characters
- min: 3.0 characters
- max: 5792.0 characters

## TOOL USAGE SEQUENCES
Average tools per question: 10.01
Maximum tools per question: 14

Most common patterns:
- rag → video_llm → vlm → rag → video_llm → vlm → rag: 64 times
- rag → vlm → video_llm → rag → video_llm → vlm → rag: 10 times
- video_llm → vlm → rag → video_llm → vlm → rag: 9 times
- rag → video_llm → video_llm → vlm → rag → video_llm → vlm → rag: 8 times
- rag → rag → video_llm → vlm → rag → video_llm → vlm → rag: 7 times
```

and settings. This combination distinguishes Ego-R1 as the first framework to simultaneously offer scalability, interpretability, and general adaptability for ultra-long video reasoning.

# G    FUTURE WORKS

Despite the possible directions emerging in the insights mentioned above, there are other possible directions to be explored based on the current dataset: **Social behavior tasks.** This dataset include different view of videos, as well as including some collaboration tasks that achieved by group of people.

**Social-behaviour analysis.**   Because the dataset contains synchronized recordings from multiple viewpoints, it can support tasks that model collaborative activities and social dynamics. For example, inferring group intentions, role allocations, or inter-person dependencies during joint tasks.

**Personal habits tracker.** A key feature of egocentric data is its tight linkage to a single, specific individual, whose routine actions reveal stable behavioral patterns. In other words, there are some special patterns for each person, for example, whether a subject brushes their teeth before or after breakfast could inform personalized reasoning models that use long-term behavioral priors to predict future actions and preferences, ultimately improving action inference accuracy.

Table 13: **System prompt for data generation.** Tool-call functions have been designed inside the AutoGen framework so that the agent is aware of how to use them.

---

[BEGIN OF GOAL]
You are an expert AI assistant specializing in analyzing human behavior and reasoning from egocentric video descriptions. You will be provided with a list of useful tools to help in reasoning the task, and your goal is to solve the user's question. The user's question is following the format: Question: <question> <timestamp> Options: <options>. You can either rely on your own capabilities or perform actions with external tools to help you. You should consider both the frequency and cost of each tool to make the best decision.
[END OF GOAL]

[BEGIN OF FORMAT INSTRUCTIONS]
When answering questions: 1. You will be provided with previous actions you have taken, based on these actions, think step-by-step about how to approach the problem. 2. Show your reasoning process clearly before providing your next action. 3. The video observation length is 10-min max. 4. For visual questions, use `video_llm` and `vlm` to explore the visual context. 5. For temporal questions, use 'rag' to explore the context before and after the event. 6. Only use the 'terminate' tool after you have thoroughly explored the question with multiple tools.
[END OF FORMAT INSTRUCTIONS]

[BEGIN OF HINTS]
1. All tools provided are crucial to the solvement of the question. You MUST exploit the usage of all tools before answering the question. 2. You may want to use the same tool multiple times with different arguments to explore the problem from different angles, if needed. 3. Make a balance between the cost and the frequency of the tools. 4. Usually, solving a question requires over 5 10 steps of reasoning, and follows a hierarchical calling structure: rag => video_llm => vlm. 5. Do not use the terminate tool too early. Instead, try to explore the question with the available tools, and only use the terminate tool when you are confident enough or have considered all the options.
[END OF HINTS]

Always structure your responses with your thought process first, followed by any tool calls. Think before you act. Think step-by-step about what information you need and which tool to use, then execute your plan exactly as reasoned without deviation. Output your thought process before using the tool, and you must strictly follow your thought process for the tool call. Currently, you are under the view of {identity}.

---

Table 14: **System prompt used during training.** We explicitly define the function-calling syntax and tool usage format, guiding the model to generate structured reasoning steps and valid tool calls that align with CoTT.

**INSTRUCTIONS**

Answer the given question. You must conduct reasoning inside <think> and </think> first every time before you get new information. After reasoning, if you find you lack some knowledge, you can call a tool from [rag, video_llm, vlm] by <tool> query </tool> and it will return the information between <information> and </information>. You can use tools as many times as your want. If you find no further external knowledge needed, you can provide the answer inside <answer> and </answer> after another thinking.
The tools you can use are:

```
{
    "name": "rag",
    "description": "Use this tool to search for information in the RAG database.",
    "arguments": {
        "type": "object",
        "properties": {
            "level": {
                "type": "str",
                "description": "The granularity of the search, choose from week|day|hour"
            },
            "keywords": {
                "type": "List[str]",
                "description": "The keywords to search for in the RAG database."
            },
            "start_time": {
                "type": "str",
                "description": "The timestamp of the start time of the search. The format should be
                    DAYX_HHMMSSFF (X is the day number, HHMMSS is the hour, minute, second, and FF is the
                    frame number(00~19))."
            },
            "query_time": {
                "type": "str",
                "description": "The timestamp of the query that was proposed by the user."
            }
        },
        "required": ["level", "keywords", "start_time", "query_time"]
    }
}
{
    "name": "video_llm",
    "description": "Use this tool to get the answer from the video language model.",
    "arguments": {
        "type": "object",
        "properties": {
            "question": {
                "type": "str",
                "description": "The question you want to use the video language model to answer."
            },
            "range": {
                "type": "str",
                "description": "The timestamp range of the video to answer the question. Use the format '
                    DAYX_HHMMSSFF-DAYX_HHMMSSFF'. The ending timestamp should be strictly larger than the
                    start timestamp. The length of the range should be smaller than 10 minutes, greater
                    than 1 second."
            }
        },
        "required": ["question", "range"]
    }
}
{
    "name": "vlm",
    "description": "Use this tool to get the answer from the vision language model.",
    "arguments": {
        "type": "object",
        "properties": {
            "question": {
                "type": "str",
                "description": "The question you want to use the vision language model to answer."
            },
            "timestamp": {
                "type": "str",
                "description": "The timestamp of the video to answer the question."
            }
        },
        "required": ["question", "timestamp"]
    }
}
```

For example, if you want to search for information in the RAG database, you can use the following tool:

```
<tool>
{
    "name": "rag",
    "arguments": {
        "level": "day",
        "keywords": ["screwdriver", "applause"],
        "start_time": "DAY1_11210217",
        "query_time": "DAY1_11220217"
    }
}
</tool>
```

