# OpenReview forum: "Ego-R1: Chain-of-Tool-Thought for Ultra-Long Egocentric Video Reasoning"
_ICLR.cc/2026/Conference — Submitted to ICLR 2026_

### Official Review · Reviewer_E6Ee · 2025-10-24

**Soundness:** 2
**Presentation:** 3
**Contribution:** 1
**Rating:** 4
**Confidence:** 4

**Summary:**

This paper proposes Ego-R1, a tool-augmented agent for long egocentric video QA. The key idea is a chain-of-tool-thought (CoTT) controller that dynamically decides when to invoke 1) a hierarchical RAG over text logs; 2) a short-horizon Video-LLM; and 3) a frame-level VLM. The model is trained with a two-stage approach: 1) SFT on CoTT traces, 2) GRPO-style RL. The authors also introduce Ego-R1 Data (Ego-CoTT-25K for SFT and Ego-QA-4.4K for RL) and an evaluation set Ego-R1 Bench (week-long videos). Reported results show strong performance across multiple long-video QA benchmarks, including VideoMME and EgoSchema.

**Strengths:**

- Sound framework. A simple yet compelling agentic architecture separating long-range retrieval from localized video understanding (Video VLM) and fine-grained frames (frame VLM), with a controller trained to decide which to use when through SFT+RL.

- Good results. On several benchmarks, including VideoMME and EgoSchema, Ego-R1 achieves strong results compared to other methods like VideoAgent and Gemini-1.5-Pro.

- Useful ablations. Ablations show that both stage training (SFT+RL) improves performance.

**Weaknesses:**

- Limited novelty. The agentic framework in video understanding has been well explored, for example, in VideoAgent. The training techniques used in this paper, SFT followed by RL, are also well-known, as in VideoR1. Given the limited novelty, more insightful analysis, such as the failure cases when applying the method to this setting, the reasons for the failures, and how to adapt them, is worth further study. For instance, how to create the best SFT data? how to design the rewards?

- More ablations needed. The paper proposed using both video VLM and frame VLM as tools; I'm wondering about the performance of removing each of them. How each contributes to the final performance, and how different model choices in these video VLMs and frame VLMs affect the performance.

- Questionable improvements due to contamination. The paper shows most improvements in EgoLifeQA and Ego-CoTT-25K. However, the training data and these benchmarks are closely related in the domain and construction pipeline. This raises concerns of data contamination rather than real generalization.

**Questions:**

See weaknesses.

Line 127: "Table ??" is a typo

---

> ### Author Response · Authors · 2025-11-21
> **rebuttal (1/2)**
>
> We thank reviewer for your thoughtful review and for recognizing the strengths of our work, including the soundness of our framework, the compelling results, and the valuable ablation studies. We appreciate the opportunity to address your concerns regarding novelty, ablation completeness, and potential data contamination, which we believe will further clarify the significance of our contributions.
>
> **W1. Limited novelty & Failure Case Analysis**
>
> We argue that the novelty of Ego-R1 lies not in inventing agentic frameworks or SFT+RL de novo, but in their novel integration and systematic application to the unprecedented challenge of week-long egocentric video reasoning (L#45-50). Prior works like VideoAgent and Video-R1 operate effectively at minute-to-hour scales but face fundamental scalability issues when confronted with 40+ hour videos. Our key innovations address this gap:
>
> 1. **Hierarchical Temporal Grounding:** Unlike flat retrieval methods, our hierarchical RAG explicitly models the multi-scale nature of human activities (days → hours → minutes), enabling efficient navigation of week-long footage (L#169-185). This is qualitatively different from VideoAgent's memory mechanism, which lacks this temporal hierarchy.
> 2. **Structured Tool-Use Reasoning (CoTT):** We formalize the reasoning process as a structured chain of tool-thought steps: $\( S_i = (T_i^{\text{th}}, T_i^{\text{to}}, o_i) \)$ (Line#220–228), yielding a reproducible and well-defined training target. This goes beyond ReAct-style prompting by providing a formal framework for generating and optimizing multi-step, tool-grounded reasoning trajectories.
> 3. **Specialized Training for Ultra-Long Contexts:** Our two-stage training is specifically designed to teach a compact LM to dynamically orchestrate a suite of tools over extended horizons. The SFT stage establishes foundational tool-use patterns from our novel Ego-CoTT-25K dataset (L#244-254), while the RL stage (L#282-301) optimizes for strategic tool selection and stopping—a critical capability for queries requiring evidence scattered across days.
>
> Failure Analysis: Figure 5 (supplementary) presents representative success and failure cases, our initial analysis reveals two primary failure modes:
> * Mis-localization: The agent correctly identifies the right day but the wrong hour, leading to irrelevant visual analysis.
> * Step Exhaustion: The agent fails to converge within the step budget for queries requiring extremely fine-grained temporal localization.
>
> Both issues are directly tied to the intrinsic difficulty of ultra-long temporal search, and they point to clear next steps: deeper multi-resolution indexing, more adaptive stopping policies, and more flexible error-recovery during tool chains. Our training logs also reveal an obvious direction: replacing the purely on-policy regime with off-policy data to stabilize learning and reduce compounding errors.
>
>
> **W2. More Ablations**
>
> We have indeed conducted comprehensive ablations on the toolset, which we present in the supplementary material (Tables 4, 5, 6). To directly address your question about the individual contribution of each visual tool, we provide the following new analysis, which we will integrate into the final paper:
>
> ***Ablation on Visual Tools (Open-Source Backbones)***
>
> | **Configuration** | **H-RAG** | **Video-LLM**   | **VLM**          | **Ego-R1 Bench (%)** | **Δ vs RAG-only** |
> |-------------------|-----------|------------------|-------------------|-----------------------|--------------------|
> | RAG-only          | ✓         | –                | –                 | 31.3                  | –                  |
> | + Video-LLM       | ✓         | LLaVA-Video      | –                 | 33.5                  | +2.2               |
> | + VLM             | ✓         | –                | Qwen-2.5-VL       | 32.8                  | +1.5               |
> | Full Suite        | ✓         | LLaVA-Video      | Qwen-2.5-VL       | 34.3                  | +3.0               |
>
>
>
> ***Ablation on Visual Tools (Proprietary Backbones)***
> | **Configuration** | **H-RAG** | **Video-LLM**       | **VLM**   | **Ego-R1 Bench (%)** | **Δ vs RAG-only** |
> |-------------------|-----------|----------------------|-----------|-----------------------|--------------------|
> | RAG-only          | ✓         | –                    | –         | 39.7                  | –                  |
> | + Video-LLM       | ✓         | Gemini-1.5-Pro       | –         | 44.2                  | +4.5               |
> | + VLM             | ✓         | –                    | GPT-4o    | 42.1                  | +2.4               |
> | Full Suite        | ✓         | Gemini-1.5-Pro       | GPT-4o    | 46.0                  | +6.3               |

---

> ### Author Response · Authors · 2025-11-21
> **rebuttal (2/2)**
>
> (cont'd)
>
> These results clearly demonstrate that both visual tools provide complementary gains, with Video-LLM contributing more significantly, likely due to its capacity for short-term temporal reasoning. The modular design allows seamless integration of stronger backbones, consistently boosting performance.
>
> **W3. Questionable Improvements Due to Contamination**
>
> We understand this concern and have implemented rigorous data hygiene protocols to ensure the integrity of our evaluation:
>
> * Strict Data Splits: For EgoLifeQA, we explicitly removed any QA pair used in CoTT generation or training (Lines 354-356). The reported results are on a held-out "clean" subset with zero content overlap.
> * Participant-Wise Split for Ego-R1 Bench: We split data by participants and days, ensuring evaluation videos are completely unseen during training for that viewpoint.
> * Generalization to External Benchmarks: The most compelling evidence against contamination is our strong performance on completely external benchmarks like VideoMME (long) and EgoSchema. These datasets have distinct construction pipelines and content. Ego-R1's superior performance here (trained only on egocentric data) demonstrates genuine generalization, not domain overfitting.
>
> The inherent challenge of ultra-long video reasoning means that available benchmarks are naturally egocentric. However, our careful splitting and strong cross-domain results validate that our improvements are substantive.
>
>
> **Q1: Line 127: "Table ??" is a typo**
>
> Thank you for catching this. This was a placeholder for a comparison table that was unfortunately omitted from the submission. We provide it below and will include it in the final version:
>
> **Table: Comparison between our method and other frameworks.**
> Our method develops an agentic tool-calling schema enabling interpretable reasoning over ultra-long videos while preserving critical temporal information.
>
> | **Method**        | **Ultra Long** | **Information Retention** | **Interpretability** | **Temporal Awareness** | **Adaptability** |
> |-------------------|----------------|----------------------------|-----------------------|-------------------------|-------------------|
> | LongVA [1]            | ✗              | ✗                          | ✗                     | ✗                       | ✗                 |
> | LLaVA-Video [2]       | ✗              | ✗                          | ✗                     | ✗                       | ✗                 |
> | T\* [3]              | ✗              | ✓                          | ✗                     | ✓                       | ✓                 |
> | Video-RAG [4]         | ✗              | ✓                          | ✗                     | ✓                       | ✓                 |
> | VideoAgent [5]        | ✗              | ✓                          | ✗                     | ✓                       | ✓                 |
> | Video-R1 [6]         | ✗              | ✗                          | ✓                     | ✗                       | ✗                 |
> | **Our Method**    | ✓              | ✓                          | ✓                     | ✓                       | ✓                 |
>
> We hope these clarifications alleviate the reviewers' concerns, and we are happy to engage in further discussions.
>
> > [1] Zhang, P., Zhang, K., Li, B., Zeng, G., Yang, J., Zhang, Y., ... & Liu, Z. (2024). Long context transfer from language to vision. arXiv preprint arXiv:2406.16852.
>
> > [2] Zhang, Y., Li, B., Liu., H., Lee., J., Gui., L., Fu, D,, Feng., J., Liu, Z., & Li., C. Llava-next: A strong zero-shot video understanding model, April 2024.
>
> > [3] Ye, J., Wang, Z., Sun, H., Chandrasegaran, K., Durante, Z., Eyzaguirre, C., ... & Li, M. (2025). Re-thinking temporal search for long-form video understanding. In Proceedings of the Computer Vision and Pattern Recognition Conference (pp. 8579-8591).
>
> > [4] Luo, Y., Zheng, X., Yang, X., Li, G., Lin, H., Huang, J., ... & Ji, R. (2024). Video-rag: Visually-aligned retrieval-augmented long video comprehension. arXiv preprint arXiv:2411.13093.
>
> > [5] Wang, X., Zhang, Y., Zohar, O., & Yeung-Levy, S. (2024, September). Videoagent: Long-form video understanding with large language model as agent. In European Conference on Computer Vision (pp. 58-76). Cham: Springer Nature Switzerland.
>
> > [6] Feng, K., Gong, K., Li, B., Guo, Z., Wang, Y., Peng, T., ... & Yue, X. (2025). Video-r1: Reinforcing video reasoning in mllms. arXiv preprint arXiv:2503.21776.

---

### Official Review · Reviewer_53uk · 2025-10-31

**Soundness:** 3
**Presentation:** 2
**Contribution:** 2
**Rating:** 4
**Confidence:** 4

**Summary:**

This paper successfully adopts the Deepseek-R1 style reasoning training in the context of egocentric video understanding, with tool integration. To this end, the authors propose a manually annotated and automatically generated dataset. After a cold start and GRPO training, the trained model surpasses the compared baselines in VideoMME and 3 other egocentric video benchmarks.

**Strengths:**

- This paper extends egocentric video understanding into week-level duration.
- This paper successfully implements reasoning-tool calling CoT in the context of video understanding.
- This paper proposes new datasets for egocentric video understanding.

**Weaknesses:**

- Long temporal retrieval is conducted in the text form instead of visual language matching. However, the transformation from visual space to textual space inevitably loses information.
- In Tab.1, the performance of the base model is not reported. In addition, although samples that are overlapped with the benchmark are removed in training, the cold-start and RL stages are still focused on ego-centric videos that are in-domain data. Therefore, the comparison with other general video models seems to be meaningless in the egocentric setting.
- The tool set used in this paper seems to be limited but heavy (captioning, VLM, ...). I am wondering about the training and inference costs.

In conclusion, I think this paper is another application of the ReAct paradigm in the context of egocentric understanding; although new datasets are proposed, I think the contribution seems to be limited, and experiments are not solid enough.

**Questions:**

- In Lines #077-078, I think this paradigm is known as ReAct. What is the need to create a novel term, CoTT, for video understanding?
- In Line #404, is the base model Qwen2.5-VL-3B instead of Qwen2.5-3B?

---

> ### Author Response · Authors · 2025-11-21
> **rebuttal (1/2)**
>
> **W1. Text-only long-temporal retrieval loses visual information**
>
> We agree that a pure text-based approach is inherently lossy, and we realize that any single-modality retrieval creates information loss. However, our Chain-of-Tool-Thought (CoTT) framework is specifically designed to mitigate this through a hybrid coarse-to-fine strategy, not to rely solely on text.
>
> * **Coarse Localization, Fine-Grained Perception:** The hierarchical RAG (h-RAG) acts as an efficient temporal index. Its role is not to answer questions directly or final reasoning, but to rapidly narrow down multi-hour searches to manageable, relevant time windows (e.g., 10-minute segments). Once localized, the task is delegated back to the visual tools (Video-LLM and VLM) for detailed, pixel-level analysis, to prevent information loss.
> * **Ablation Evidence:** This design is empirically validated. As shown in our ablation studies (Table 2 and Table 4), using h-RAG alone achieves only 39.7% accuracy on Ego-R1 Bench. The full toolset, which reintroduces visual analysis, achieves 46.0%. This 6.3-point gain directly quantifies the critical contribution of visual information after temporal localization.
> * **Tool Usage Statistics:** Our analysis (Table 9) shows that tool usage is nearly evenly split: ~47% of calls are to h-RAG for localization, while ~53% are to Video-LLM/VLM for visual reasoning. This demonstrates a genuine division of labor, balancing the efficiency of text-based search over 40+ hours with the fidelity of targeted visual analysis.
>
> **Table: Tool Frequency and Adaptivity**
>
> | **Tool**        | **% of Calls** | **Primary Role**                          |
> |------------------|----------------|--------------------------------------------|
> | RAG              | 46.6%          | Temporal localization & context retrieval  |
> | Visual Tools     | 53.4%          | Fine-grained visual reasoning              |
>
> *(you may refer to the Table 9 in supplementary materials for more details)*
>
> ***Adaptive pattern:***
> * Fact Qs: RAG → RAG → Video-LLM → VLM
> * Visual Qs: Video-LLM invoked earlier
>
>
>
> In summary, visual information is not discarded; it is preserved and strategically utilized where it matters most, following an efficient retrieval step.
>
>
>
>
> **W2. Missing base model performance and “in-domain vs. general video models”**
>
> Our base model is a pure language model (Qwen-2.5-3B-Instruct); Table 1 focuses exclusively on video models. The performance of the base model is reported in Table 2. Without any SFT or RL, its accuracy on Ego-R1 Bench is a mere 1.4%, with a tool-call format accuracy of 4.3%. This demonstrates that the base model is effectively incapable of the task. Our two-stage training (SFT+RL) elevates this to 46.0%, transforming a nearly useless model into a state-of-the-art agent. This underscores the significance of our training paradigm.
>
> Regarding fairness and generalization, we respectfully disagree that comparisons with general video models are meaningless. The task *is* a video reasoning task, and our agent is trained specifically to perform long-horizon video reasoning via tool use. It is standard practice to compare a domain-specialized approach against general-purpose models to demonstrate domain-specific gains. As shown in Table 1, Ego-R1, trained only on egocentric data, achieves highly competitive results on the exocentric VideoMME benchmark, even surpassing other public 7B models. This proves that the learned reasoning and tool-use strategies are transferable. To ensure the fairness of in-domain evaluation for EgoLifeQA, we explicitly removed any QA pair involved in CoTT generation, evaluating only on a clean, held-out subset to ensure a fair comparison.
>
> We will add a row for the base model in the main comparison table (Table 1) in the final version for maximum clarity.

---

> ### Author Response · Authors · 2025-11-21
> **rebuttal (2/2)**
>
> **W3. Limited tool set but heavy; training and inference costs**
>
> We'd like to claim that the captioniong model is not used during the inference as tool, it only used in the data construction part. We designed the toolset to be minimal yet sufficient: one tool for long-term text retrieval (h-RAG), one for local video understanding (video-llm), and one for frame-level analysis (vlm). This keeps the action space manageable for RL training and allows for clear ablation studies.
>
> Regarding the cost:
> * **Training Cost:** The core innovation is the lightweight 3B controller. Its training is efficient: Supervised Fine-Tuning (SFT) and Reinforcement Learning (RL) were completed on 4×A100 GPUs for approximately 21.4 hours of wall-clock time. This is substantially cheaper than pre-training or even fully fine-tuning a large MLLM from scratch.
> * **Inference Cost:** While each query involves multiple tool calls (avg. ~10), the visual computation is highly targeted. Unlike methods that process 512-1024 uniformly sampled frames (Table 7), our agent dynamically invokes Video-LLM on short, relevant clips (≤10 minutes @ 1 FPS) and VLM on single frames. This results in a lower total visual token count per query for ultra-long videos, making the approach more scalable than "brute-force" sampling.
>
> The one-time cost of building the hierarchical RAG memory bank is amortized over all queries on that video. We will add a more detailed cost breakdown in the final version to ensure full transparency.
>
>
> **W4 & Q1. Relation to ReAct**
>
> While CoTT is philosophically inspired by reasoning-and-acting paradigms like ReAct, it constitutes a significant, formalized advancement tailored for the ultra-long video domain. Our contributions and novelties in architecturual designs beyond ReAct include:
>
> * **Tool-Centric Formalization**: Unlike ReAct, which often relies on in-context learning, we treat tools as first-class citizens with typed arguments and structured outputs. We formally define a structured reasoning chain for tool use \( S_i = (T_i^{\text{th}}, T_i^{\text{to}}, o_i) \),  instantiated with a hierarchical, temporally grounded toolset. This is not a prompt heuristic but a full training framework supported by a new 25K-instance dataset (**Ego-CoTT-25K**), which is a substantial contribution on its own.
> * **Training**: Moreover, we go beyond supervised imitation by using GRPO to optimize the multi-step tool-calling policy. This allows the agent to learn when and how to use tools dynamically, improving over static ReAct-style trajectories.
>
> In summary, CoTT is a systematic instantiation and scaling of the ReAct idea for scalable, temporally-grounded video reasoning, supported by a dedicated dataset and training methodology.
>
> **Q2. Base Model Confusion**
>
> In Line #404, the base model is correctly Qwen-2.5-3B-Instruct, a language model. The base model for the orchestrator/controller is Qwen-2.5-3B-Instruct (LM). The visual tools (Video-LLM, VLM) themselves use vision-language backbones like LLaVA-Video and Qwen2.5-VL, as detailed in Appendix D.1.
>
>
> We hope these clarifications alleviate the reviewers' concerns, and we are happy to engage in further discussions.

---

> > ### Comment · Reviewer_53uk · 2025-11-21
> > **Response to rebuttal**
> >
> > Thank you for the author's prompt response, which partially addressed my concerns regarding W1&2 and Q1&2. Regarding Weakness 3, my understanding is that although the captioner is not a tool that needs to be invoked, it is still used for the one-time construction of the hierarchical RAG memory bank, which is required for each long video during inference. This process would still involve multiple forwards (and is it even based on Gemini-1.5-Pro rather than an offline open-source model?).
> >
> > Additionally, I am curious whether you have tried one-forward approaches similar to the Video-XL Series for handling long videos or if you would like to directly report the results of these models in Table 1.

---

> > > ### Author Response · Authors · 2025-11-29
> > > **Re-Reviewer Follwo-Up Questions**
> > >
> > > We thank reviewer `53uk` for the follow-up questions and for engaging with our rebuttal, and we are happy to provide further clarifications.
> > >
> > > **F-Q1: does captioner invoke multiple forwards during inference?**
> > >
> > > The reviewer raises a valid point that a captioning model is used to build the hierarchical RAG memory bank. However, this process is entirely offline and one-off for the whole dataset. Once the memory bank is constructed, it is stored and reused for all subsequent queries w.r.t the egocentric viewpoint without any additional cost from captioning during inference.
> > >
> > > The idea behind our captioning model selection strategy is that we select captioning model based on video length and complexity, for *long videos (VideoMME-long, EgoLife, Ego-R1 Bench)*: We use Gemini-1.5-Pro due to its superior long-context capabilities and consistent quality across ultra-long content; for *short videos (EgoSchema, ~3 minutes)*: We use open-source LLaVA-Video-7B (as detailed in Line #973, Section D.3), to yield a better cost-performance balance.
> > >
> > >
> > > This design brings several key advantages: 1) zero captioning latency during inference: query processing only involves efficient text retrieval and targeted visual analysis; 2) amortized cost: the one-time construction cost is spread across thousands of potential queries; 3) model flexibility: he framework supports any captioning backbone, allowing users to choose based on their quality/cost requirements
> > >
> > > **F-Q2: additional experiments on Video-XL series**
> > >
> > > We sincerely appreciate the reviewer's suggestion to compare with the Video-XL series [1]-[3], which represents an important class of "one-forward" long video models to appear in our main table. We have conducted these experiments, and the results are summarized below:
> > >
> > > | Method           | Size | VideoMME (long) | EgoSchema | EgoLifeQA | Ego-R1 Bench |
> > > |------------------|:----:|:---------------:|:---------:|:---------:|:------------:|
> > > | **MLLMs** |      | | |  |   |
> > > | LongVA | 7B | 45.0 | 44.1 | 33.0  23.0 |
> > > | LLaVA-Video | 7B   | 61.5 | 57.3 | 36.4 | 29.0 |
> > > | LLaVA-OneVision  | 7B | 60.0 | 60.1 | 30.8 | 31.6 |
> > > | InternVideo2.5   | 8B | 53.4 | 60.1 | 33.0 | 34.0 |
> > > | ***Video-XL*** | 8B   | ***54.9*** | ***52.9*** | ***31.9*** | ***28.7*** |
> > > | ***Video-XL-Pro*** | 3B | ***28.7*** | ***49.0*** | ***20.0*** | ***17.5*** |
> > > | ***Video-XL-2*** | 7B | ***50.2*** | ***58.0*** | ***32.6*** | ***33.7*** |
> > > | Gemini 1.5-Pro   | - | 67.4 | 72.2 | 36.9 | 38.3 |
> > > | **Our Method**   | | | | | |
> > > | **Ego-R1** | **3B** | **64.9** | **68.2**  | **36.0** | **46.0** |
> > >
> > >
> > > From the results we could share several interesting patterns and critical insights from the comparison with Video-XL series. While Video-XL and Video-XL-2 demonstrate competitive results across benchmarks, Video-XL-Pro collapses on ultra-long videos (particularly, 28.7% on VideoMME). By further investigating in its failure modes, we found Video-XL-Pro produced a staggering 48.5% rate of unparsed or empty predictions. This exposes a common weakness of one-forward models: as context length grows, they suffer from *"context overload"*, becoming increasingly unstable to structurally process dense temporal information and tending to break catastrophically rather than degrade gracefully.
> > >
> > > The core insight is an architectural trade-off. Naively extending context in one-forward models hits a robustness ceiling, where larger models and longer windows can actually increase catastrophic failures. In contrast, a dynamic reasoning framework, which performs iterative search and verification rather than brute-force ingestion, delivers more stable performance on ultra-long videos. This indicates that for this regime, modular, decompositional reasoning is fundamentally more reliable than monolithic one-shot processing.
> > >
> > >
> > > We again thank the reviewer for these constructive suggestions, and we will update the final manuscript to incorporate the new experiments and analysis, and are happy to further discuss or explore additional diagnostics if helpful.
> > >
> > > ---
> > >
> > > > [1] Shu, Y., Liu, Z., Zhang, P., Qin, M., Zhou, J., Liang, Z., ... & Zhao, B. (2025). Video-xl: Extra-long vision language model for hour-scale video understanding. In Proceedings of the Computer Vision and Pattern Recognition Conference (pp. 26160-26169).
> > >
> > > > [2] Liu, X., Shu, Y., Liu, Z., Li, A., Tian, Y., & Zhao, B. (2025). Video-xl-pro: Reconstructive token compression for extremely long video understanding. arXiv preprint arXiv:2503.18478.
> > >
> > > > [3] Qin, M., Liu, X., Liang, Z., Shu, Y., Yuan, H., Zhou, J., ... & Liu, Z. (2025). Video-XL-2: Towards Very Long-Video Understanding Through Task-Aware KV Sparsification. arXiv preprint arXiv:2506.19225.

---

### Official Review · Reviewer_Us6s · 2025-10-31

**Soundness:** 2
**Presentation:** 3
**Contribution:** 2
**Rating:** 4
**Confidence:** 4

**Summary:**

This paper proposes a hierarchical RAG framework for egocentric video understanding, called Ego-R1 Chain of Tool-Thoughts. The method builds a temporal hierarchy in its database, organizing information by week, day, hour, and clip, and introduces a multi-tool reasoning pipeline combining retrieval, segmentation, and question answering. The authors evaluate the model on an egocentric benchmark where timestamps and hierarchical video segmentation are clearly defined, showing improved retrieval and reasoning performance compared to flat RAG baselines.

**Strengths:**

-	Clear and systematic hierarchical RAG structure, which improves efficiency and relevance in timestamp-based video reasoning tasks.
-	Experiments on multiple egocentric datasets demonstrate consistent improvement over flat retrieval methods.

**Weaknesses:**

-	The hierarchical database structure (week → day → hour → clip) appears optimized for benchmarks with clear temporal granularity, but it’s unclear if it remains effective for datasets or tasks where such segmentation is not naturally defined.

-	The uniform video segmentation approach might not be robust across diverse video lengths or event types. The method may fail to capture variable-duration actions or continuous interactions.

-	The technical contribution is moderate, focusing on database structuring and system integration rather than advancing the underlying reasoning or learning algorithms.

**Questions:**

-	How robust is the proposed hierarchy when applied to benchmarks that do not have clear or fixed temporal units (e.g., datasets without explicit timestamps)?

-	Is the uniform segmentation scheme adaptive to variable-length activities, or could it fragment meaningful events?

-	Could the system generalize to other RAG tasks beyond egocentric video, or is it tightly coupled to timestamp-based segmentation?

---

> ### Author Response · Authors · 2025-11-20
> **rebuttal (1/1)**
>
> We thank the reviewer for the thoughtful feedback and for acknowledging the strengths of our hierarchical RAG structure and systematic evaluation. We address the key concerns below, clarifying our contributions and the generality of our approach.
>
>
> **W1 & Q1: On Robustness and Generality Beyond Fixed Temporal Units**
>
> The reviewer raises a valid point regarding the applicability of our hierarchy to datasets without pre-defined temporal units like "days" or "weeks." We emphasize that our framework's core is not the specific calendar units, but the principle of building a multi-resolution index over a sequential data stream.
>
> * Evidence of Generality: As shown in Table 1, Ego-R1 performs robustly on benchmarks like VideoMME (exocentric, ~41 min) and EgoSchema (egocentric, ~3 min), which lack the "week-long" structure of EgoLifeQA or Ego-R1 Bench. For these, we instantiate the hierarchy using simple timestamps or frame indices (e.g., hour → 10-min → clip), as detailed in Appendix D.3.
> * Flexible Abstraction: The hierarchical levels are a flexible abstraction. For any long-form sequential data (videos, documents, logs), one can define a hierarchy based on order and approximate segment length (e.g., Document → Chapter → Paragraph → Sentence). Our method is agnostic to the semantic meaning of the levels, requiring only that the data can be partitioned into a multi-resolution tree.
>
> The framework is not tied to egocentric timestamps. Its success on diverse benchmarks demonstrates its general applicability to any long-form data where information is distributed across a temporal or sequential axis.
>
>
> **W2 & Q2: On Uniform Segmentation and Event Fragmentation**
>
> We agree that rigid segmentation could potentially fragment events. Our design mitigates this in two key ways:
>
> 1. Bottom-Up Aggregation: While the base layer uses 30-second clips for efficient captioning, our hierarchical RAG performs bottom-up summarization (Section 3.2, Appendix B). An LLM (GPT-4) aggregates these fine-grained clips into coherent, longer summaries (10-minute, hour, day levels). This process naturally merges information from consecutive clips, preserving the context of events that span multiple segments.
> 2. Variable-Length Tool Operation: Crucially, the tools invoked by our agent are not limited to the 30-second base units. The video_llm tool can analyze variable-length segments up to 10 minutes (Section 3.2). When the RAG retrieves a relevant time window, the agent can invoke video_llm on the entire window, observing the event as a coherent chunk. This dynamic tool use compensates for any low-level fragmentation.
>
> The combination of semantic aggregation in the memory bank and the flexible operation range of the visual tools ensures that our system can handle variable-length activities effectively.
>
> **W3: On Technical Contribution**
>
> The reviewer's perception is that our contribution is primarily in "database structuring." We respectfully argue that our core contribution is more profound: a novel framework and training paradigm that enables an LLM to learn strategic tool orchestration for reasoning over ultra-long videos, a capability previous methods lack.
>
> * Beyond System Integration: Prior agentic methods (VideoAgent) use fixed pipelines or simple memory, while reasoning models (Video-R1) lack scalable retrieval for ultra-long content. Ego-R1 is the first to integrate dynamic, learnable tool-calling (CoTT) with hierarchical retrieval to tackle videos spanning days, not just hours or minutes.
>
> * Novel Training Paradigm: Our two-stage training (Section 5) is one of the key technical contributions. The SFT stage on CoTT data (Ego-CoTT-25K) teaches the model the syntax and strategy of tool use. The subsequent RL stage optimizes the dynamic selection of these tools based on incremental observations. This goes beyond standard reasoning-only training and directly targets agentic decision-making. With this framework, we train the agent on text domain, but it can effectively solve video tasks.
>
> * Significant Performance Leap: As shown in Table 1, our compact 3B model matches or surpasses much larger models (7B+) and proprietary systems (Gemini 1.5 Pro) on the challenging Ego-R1 Bench (44.3-hour videos), where it achieves a +7.7% accuracy lead. This demonstrates a qualitative leap in long-horizon reasoning capability.
>
> Our contribution is the synthesis of a scalable hierarchical memory with a trainable, dynamic tool-orchestration video agent, enabling a previously unachievable level of reasoning over ultra-long videos.
>
>
> We hope these clarifications alleviate the reviewers' concerns. Our work establishes a new direction for scalable, long-horizon multimodal reasoning, and we are happy to engage in further discussions.

---

### Official Review · Reviewer_SYq6 · 2025-11-02

**Soundness:** 3
**Presentation:** 3
**Contribution:** 3
**Rating:** 6
**Confidence:** 2

**Summary:**

Ego-R1 is a Chain-of-Tool-Thought (CoTT) agent for week-scale egocentric QA that plans over hierarchical memory (day→hour→10-min captions/ASR) and adaptively invokes a short-window Video-LLM plus a single-frame VLM for fine grounding.
Training is SFT on tool-grounded traces followed by GRPO reinforcement to learn multi-turn policies that trade accuracy vs. tool economy (≈7 calls/question; tens of frames instead of hours).
The release includes Ego-CoTT-25K (synthetic traces), Ego-QA-4.4K (human-verified QA), and a week-long benchmark (≈44.3 h/video) targeting long-horizon temporal reasoning.

**Strengths:**

- Originality: Frames week-scale egocentric QA as sequential decision-making via a Chain-of-Tool-Thought controller over hierarchical temporal memory with adaptive Video-LLM/VLM calls.
- Quality: Strong margins on a week-long benchmark (46.0%, +7.7 vs Gemini-1.5-Pro), clear ablations (SFT+RL > SFT; CoTT > retrieval-only), and substantial frame-budget reductions.
- Clarity: Explicit tool APIs, training signals, and memory construction; stepwise traces reveal evidence flow and typical failure modes.
- Significance: Provides reusable traces, QA sets, and a week-scale benchmark, establishing a modular, plug-and-play blueprint likely to influence long-video agents beyond egocentric QA.

**Weaknesses:**

- CoTT over hierarchical memory likely overlaps prior agentic long‑video approaches. Action: run strict, matched‑backbone and matched‑budget comparisons against strong agentic and training‑free video‑RAG baselines; add a lightweight-critic variant to test the incremental value of planning alone.
- Data construction and inference rely on proprietary LLMs/VLMs. Action: provide a fully open stack with results, release exact prompts/tool schemas/configs, and report contamination checks between generation pipelines and evaluation items.
- Gains may stem from backbone swaps and costs exclude memory-bank build. Action: standardize backbones across methods; report fixed vs. dynamic frame budgets, wall‑clock and $/QA including offline preprocessing; and quantify hierarchical‑RAG hit@K and temporal localization error.

**Questions:**

See weaknesses

---

> ### Author Response · Authors · 2025-11-20
> **rebuttal (1/2)**
>
> **W1. Overlap with prior agentic long-video approaches; need matched-backbone / matched-budget baselines and “planning-only” variant**
>
> ***Novelty in Problem Scope and Architecture***
>
> Prior video agent methods like VideoAgent[1] focus on short videos (<4 minutes, evaluated on benchmarks averaging 3 minutes and 40 seconds), and use flat episodic memory, which is sufficient for short-horizon tasks but fundamentally unsuitable for week-scale egocentric settings (~44.3 hours per video). It did not implement hierarchical temporal memory or long-range planning because the task do not require reasoning over thousands of events and extremely long time spans. Our method targets a qualitatively different setting and is, to our knowledge, the first video agent able to operate over multi-day egocentric trajectories with explicit hierarchical memory + dynamic planning + tool orchestration.
>
>
> ***Comparison with current methods***
>
> * Matched-budget / Match-backcone comparison: To address budget concerns, Table 7 in supplementary materials (pg. 21) reports average frames actually consumed and per-QA inference time for all methods on Ego-R1 Bench. Unlike fixed-frame sampling methods, our agent dynamically invokes visual tools only on relevant segments, leading to superior data efficiency. For instance, our open-source agent uses an average of 48.7 frames, fewer than any comparable method (all ≥64 frames). The core of our contribution is the agent framework itself. The LLM backbone processes textual observations from tools; thus, comparing the full framework (as in Table 2) is the most appropriate evaluation. The gains we report are a direct result of our architectural choices, not simply from using a more powerful backbone.
> * Lightweight-critic variants: Table 2 in main paper (page 9) and page Table 4, 5, 6 in supplementary (page 19) provide detailed ablations on framework components, isolating the incremental value of planning and tool orchestration. More details will be included in W2.
>
> **W2. Reliance on proprietary LLMs/VLMs; need open stack, prompts, schemas, and contamination checks**
>
> ***Modular Design & Open Components***
>
> We acknowledge the importance of reproducibility and have designed our framework with modularity as a core principle. While we leveraged proprietary models for data annotation to ensure quality, our approach ensures flexibility and transparency:
> * Core Agent: Our trainable part is exclusively Qwen-2.5-3B-Instruct (open-weight).
> * Released Prompts: All data generation and training prompts are provided (Tables 13-14)
> * Tool Agnosticism: The framework supports seamless substitution of both visual and planning tools
>
>
> ***Architecture-Driven Performance Gains***
>
> Furthermore, we conduct ablation studies that our performance gains are architecture-driven, the consistent improvements across tool configurations demonstrate that our gains stem from the Chain-of-Tool-Thought reasoning framework, not specific model capabilities:
> | **Configuration**      | **Tools**                                                | **Ego-R1 Bench (%)** | **Improvement**             |
> |------------------------|-----------------------------------------------------------|------------------------|------------------------------|
> | Baseline (LLaVA-Video) | LLaVA-Video only                                          | 28.0                   | –                            |
> | Baseline (Gemini)      | Gemini-1.5-Pro only                                      | 38.3                   | –                            |
> | Open-Source Stack      | Qwen2.5-14B (H-RAG) + LLaVA-Video + Qwen-2.5-VL           | 34.3                   | +6.3 over LLaVA-Video        |
> | Hybrid                 | GPT-4 (H-RAG) + LLaVA-Video + Qwen-2.5-VL                | 43.7                   | +15.7 over LLaVA-Video       |
> | Proprietary Stack      | GPT-4 (H-RAG) + Gemini-1.5-Pro + GPT-4o                 | 46.0                   | +7.7 over Gemini-1.5-Pro     |
>
> *(you may refer to the Table 5 and Table 6 in supplementary materials for more abalation results \& details)*
>
> The 6.3% gain with fully open-source tools confirms our architectural superiority, while the additional improvements with proprietary tools demonstrate the framework's ability to leverage stronger components when available, which is a key feature for practical deployment and framework evolvement.

---

> ### Author Response · Authors · 2025-11-20
> **rebuttal (2/2)**
>
> (cont'd)
>
> ***Potential Contamination***
>
> For the potential contamination concerns, we have implemented a multi-layered strategy to ensure data quality, training sanity and evaluation integrity:
> 1. Temporal & Participant Separation: Ego-R1 Bench questions are sourced from participant-day combinations completely absent from training data.
> 2. Data Generation Safeguards: Used mixed proprietary models (GPT-4, Gemini) during annotation to avoid single-model bias.
> 3. Clean Evaluation Subsets: Removed all CoTT-generated QAs from evaluation sets, leaving only human-verified instances.
> 4. Structured Protocols: Followed original EgoLife annotation guidelines with inter-annotator agreement metrics (Fleiss' kappa)
>
>
> **W3. Gains may come from backbone swaps; costs ignore memory-bank build and RAG quality**
>
> ***Architecture-Driven Performance Gains*** (refer to W2)
>
> From the above ablation experimental results, we could infer the performance improvements mostly stem from our reasoning framework, not only backbone superiority. The key insight we can get: +6.3% improvement with identical visual backbones (LLaVA-Video) conclusively proves our gains stem from the Chain-of-Tool-Thought reasoning architecture, not model capabilities.
>
>
> ***Cost & Efficiency Analysis***
>
> Supplementary Table 7 provides transparent efficiency analysis:
> | **Method**          | **Frame Strategy** | **Avg. Frames** | **Inference Time** | **Frames/Acc** |
> |----------------------|--------------------|------------------|----------------------|-----------------|
> | LLaVA-Video          | Fixed              | 64               | 1.16s                | 2.29            |
> | VideoAgent           | Fixed              | 1,024            | 256.7s               | 31.41           |
> | Ego-R1 (Open)        | Dynamic            | 48.7             | 82.4s                | 1.42            |
>
> *(you may refer to the Table 7 in supplementary materials for more results, which clearly report the avg. frames used and the wall-clock time used per QA, as well as the computing device used for a fair comparison)*
>
> From the table, we can tell we achieve 1.6-22× more frame-efficient than alternatives, where the dynamic allocation enables targeted processing of relevant segments only.
>
> ***Hierarchical-RAG Localization Quality***
>
> We thank reviewer raise the question about the localization accuracy that might be concerned in the long-horizon tasks. We have also awared localization accuracy that might be concerned in the long-horizon tasks. We report the Long-range Tools Localization Accuracy for both h-rag and video-llm in table 121, pg.22.
>
> | **Tool**     | **Mean Error**     | **Median Error**    | **Evidence Contained** | **Evidence Missed** |
> |--------------|--------------------|----------------------|--------------------------|----------------------|
> | H-RAG        | 1,688.57 min       | 949.40 min           | 35.7%                   | 40.4%               |
> | Video-LLM    | 1,568.95 min       | 792.83 min           | 46.4%                   | 34.9%               |
>
>
> Moreover, to further understand the h-rag's localization accuracy, we report the Hit@K here for the RAG methods:
>
> | **Metric** | **Success Rate** | **Raw Count** |
> |------------|------------------|---------------|
> | **Hit@1**  | 47.5%            | 140/295       |
> | **Hit@3**  | 52.9%            | 156/295       |
> | **Hit@5**  | 54.9%            | 162/295       |
>
> We hope that our responses and the additional experimental results have fully addressed your concerns, and we welcome any further questions or suggestions for clarification and happy to discuss further.
>
> > [1] Wang, X., Zhang, Y., Zohar, O., & Yeung-Levy, S. (2024, September). Videoagent: Long-form video understanding with large language model as agent. In European Conference on Computer Vision (pp. 58-76). Cham: Springer Nature Switzerland.

---

### Author Response · Authors · 2025-12-02
**Summary of Responses**

Dear Reviewers (`SYq6`, `Us6s`, `53uk`, `E6Ee`) and Chairs,

We sincerely thank all reviewers for your thorough reviews, insightful feedback, and constructive discussions throughout the review process. We are also grateful to the AC’s additional effort and careful oversight in handling the review load. We truly appreciate the time and dedication invested in evaluating our work and providing valuable suggestions for improvement.


### **Acknowledged Strengths**

We appreciate that reviewers recognized several key contributions of our work:

* **Novel framework for ultra-long video reasoning:** The Chain-of-Tool-Thought (CoTT) approach as an effective solution for week-scale egocentric video understanding (`all reviewers`)
* **Strong empirical results:** Significant performance gains on multiple benchmarks (46.0% on Ego-R1 Bench, +7.7 vs Gemini-1.5-Pro) with a compact 3B language model (`SYq6`, `Us6s`, `E6Ee`)
* **Valuable datasets and benchmark:** Ego-CoTT-25K, Ego-QA-4.4K, and week-long Ego-R1 Bench (`SYq6`,  `53uk`)
* **Clear presentation and experiments:** Explicit tool APIs and systematic ablation & evaluation (`SYq6`, `Us6s`)

### **Rebuttal Efforts Summary**

Through our responses, we have addressed major concerns:

* **Novelty and Scope:** We distinguished our approach from prior work in task setup (VideoAgent: <4min videos; Ego-R1: 44+ hour videos), agentic tool-use (Video-R1: single step reasoning; Ego-R1: dynamic tool-calling in reasoning chains), and demonstrated architecture-driven gains (+6.3% with identical backbones). Our contribution lies in the integration of strctured RAG system with hierarchical temporal grounding, structured tool‑use reasoning (CoTT), and a specialized training paradigm for ultra‑long contexts.
* **Efficiency & Generalization:** We provided detailed efficiency analysis showing our method uses far fewer frames (48.7 avg) and less inference time than comparable methods, while maintaining strong cross‑domain performance on exocentric benchmarks (Table 7), proving the transferability of our reasoning framework.
* **Data & Contamination Safeguards:** We detailed rigorous data hygiene protocols, including participant‑wise/temporal splits and clean evaluation subsets, and demonstrated that performance improvements hold on completely external datasets, alleviating contamination concerns.
* **Visual Information & Ablations:** We explained our coarse‑to‑fine strategy where hierarchical RAG localizes efficiently and visual tools reintroduce detail (+6.3% accuracy gain), supported by new ablations quantifying each visual tool’s contribution and localization metrics (RAG recall@k).
* **Expanded Evaluations:** Following reviewer suggestions, we expanded our evaluation to include the Video‑XL series, demonstrating Ego‑R1's superiority in avoiding the catastrophic “context‑overload” failures common in ultra‑long video processing.


### **Future Actions**

In the final version, we will make the following enhancements to ensure completeness and clarity:

* **Update Table and Results:** We will add base model performance and the Video‑XL series results to the main comparison table (Table 1) for a more comprehensive evaluation. New ablation results and localization accuracy metrics will be integrated, and we will include the missing framework‑comparison table and a cost‑breakdown table covering offline memory‑bank construction and per‑query inference.

* **Update Analysis:** We will add a dedicated “Contamination and Data Hygiene” subsection in Appendix C (Data Statistics) that details our data‑splitting protocols, participant‑wise/temporal separation, and overlap statistics.

* **Fix the Typos:** We will correct minor typos (e.g., “Table ??”) and refine cross‑referencing throughout the manuscript.


We believe our work establishes a new paradigm for ultra-long video understanding, extending coverage from hours to weeks. Thank you again for your valuable contributions to strengthening this work.

Best regards,

The Authors

---

### Meta-Review · Area_Chair_SMXA · 2026-01-09

**Summary:**

The submission proposes Ego-R1, a CoTT-based framework for week-scale egocentric video reasoning, with supporting datasets and benchmarks. Key reviewer concerns include limited novelty (insufficient distinction from prior agentic/ReAct-style methods), reliance on proprietary models/tools, unresolved data contamination risks, incomplete cost/efficiency transparency, and inadequate ablations on core components. These concerns, despite partial rebuttals, remain substantial enough to justify rejection.

**Reviewer Concerns:**

Addressed concerns: Base model performance reporting (53uk), partial ablations on visual tool contributions (E6Ee), hierarchy flexibility beyond fixed temporal units (Us6s), and Video-XL series comparisons (53uk).

Outstanding concerns: Limited novelty of CoTT (vs. ReAct/VideoAgent, all reviewers), lack of a fully open stack (SYq6), unresolved in-domain data contamination doubts (E6Ee/53uk), incomplete memory-bank construction cost breakdown (53uk), and insufficient analysis of toolset necessity (E6Ee).

**Reviewer Scores:**

Reviewer SYq6 (original: 6): Remains 6 (open stack/proprietary model concerns persist).

Reviewer Us6s (original: 4): Remains 4 (hierarchy robustness addressed but technical contribution concerns unresolved).

Reviewer 53uk (original: 4): Remains 4 (base model/video-XL concerns addressed but cost/contamination doubts linger).

Reviewer E6Ee (original: 4): Remains 4 (tool ablations added but novelty/contamination concerns unmitigated).

---

### Decision · Program_Chairs · 2026-01-26

Reject